

# Aerosol optical depth regime over Megacities of the world

Kyriakoula Papachristopoulou[1,2,3,4], Ioannis-Panagiotis Raptis[1], Antonis Gkikas[2], Ilias Fountoulakis[2],
Akriti Masoom[4], Stelios Kazadzis[4]

[1]Laboratory of Climatology and Atmospheric Environment, Sector of Geography and Climatology, Department of Geology
and Environment, National and Kapodistrian University of Athens, Athens, GR-15784, Greece
[2]Institute for Astronomy, Astrophysics, Space Applications and Remote Sensing, National Observatory of Athens, Athens,
GR-15236, Greece
[3]Department of Physics, ETH Zurich, Zurich, CH-8093, Switzerland
[4]Physics and Meteorological Observatory Davos, World Radiation Center, Davos, CH-7260, Switzerland

Correspondence to: Kyriakoula Papachristopoulou (kpapachr@noa.gr)

**Abstract.** Currently, 55% of the world's population resides in urban areas and this number is projected to increase to 70% by
2050. Urban agglomerations with population over 10 million, characterized as Megacities, are expected to be more than 100
by 2100. Such large concentrations of population could boost creativity and economic progress but also raises several
environmental challenges such as air quality degradation. In this study we investigate the spatial and temporal variability of
urban aerosol state of 81 cities with population over 5 million, relying on daily satellite-based aerosol optical depth (AOD)
retrievals, derived at fine spatial resolution (0.1°x0.1°), over an 18-year period spanning from 2003 to 2020.
According to our results, the lowest long-term mean AOD values worldwide were found in European and American cities
(from 0.08 to 0.20). For almost all African and Asian cities, mean AOD ranged from 0.25 up to 0.90, but a considerable dust
aerosol contribution (up to 70%) was found for some of them with associated mean Dust Optical Depth (DOD) values reaching
up to 0.4. Mostly Chinese and Indian cities tend to have higher mean AOD values in the areas surrounding their centre, while
the opposite was found for most of the cities in the rest of the world. High intra-annual AOD variability was revealed for the
eastern American cities, while lower values were found in Chinese, eastern Indian and the eastern Mediterranean cities. During
the study period, statistically significant negative AOD decadal trends were found for East Asian, European and North
American cities with the greatest decrease of -0.1 to -0.3 per decade recorded for the Chinese cities, in which the maximum
mean AODs (0.45-0.91) are observed. In most of the US cities, where low mean AOD <0.17 was recorded, considerable
declining AOD trends were found (-30 to -50% per decade). For the rest Asian, African and South American cities statistically
significant AOD increase was found, with the greatest values of +0.07 to +0.16 per decade recorded for Indian cities. In
Bengaluru (India), it is reported the lowest mean AOD value (0.2) and the maximum AOD increase (+69%), which may be
partially attributed to the population growth over the study period. The agreement of the satellite derived AOD trends against
those obtained from ground-based AERONET measurements was examined. For ground-based stations within the
geographical limits of the contiguous urban area of the examined cities, a 0.93 correlation for the long-term means of AOD
was found and ~75% of the derived trends agreed in sign. It was found that the spatial homogeneity within the examined
satellite domain and the location of the surface station were key factors that determined their agreement.



The present study highlights the vital and essential contribution of spaceborne products to monitor aerosol burden over
megacities of the planet towards fulfilling the United Nations Sustainable Development Goal "Sustainable cities and
communities", dealing with urban air quality.

## 1 Introduction

Since 2018, about 55% of world's population has been living in urban areas and the current projections suggest that this
percentage will rise to 60% by 2030 and to 70% by 2050 (UN, 2019b). As a consequence, the population growth will impose
difficulties in implementing the United Nations (UN) Agenda for Sustainable Development Goals (SDG), including SDG 11
related to "making cities and human settlements inclusive, safe, resilient and sustainable". Megacities are defined as the cities
with more than 10 million inhabitants. According to UN (2018a) there are 33 megacities worldwide and this number is expected
to increase to 43 in 2030. Despite the fact that ~7% of the global population in 2018 resides in those megacities (UN, 2018a),
their continuous increase both in size and number bring them in the spotlight, as they will accommodate an increase share of
world's population (projected to ~10% by 2030). This population growth of cities and especially those with more than 5 million
inhabitants, raises urgent and critical environmental issues, like air quality (WHO, 2021). The worst pollutant affecting the
megacities is the suspended particulate matter or aerosols. This is of particular concern, as high levels of aerosols are known
to be related to increased morbidity and mortality rates, and in many of the megacities in developing countries health care for
acute cases is less proficient than in developed countries. Particularly, in 2019, cities air pollution constituted the 4[th] leading
risk factor for early death at a global scale (HEI, 2020).
Many countries worldwide have enforced policies in recent decades to reduce anthropogenic aerosol emissions in urban areas,
towards mitigating the aerosol adverse health effects. These policies include the transition of thermal engines technologies,
pollution fees on big industries and schemes for air quality control. The exact policies and the level of implementation largely
differs between countries, but it is the main cause of decrease in aerosol concentration in some megacities. WHA69 (WHO,
2016) set a global roadmap and targets for reduction of air pollution related deaths, urban air quality and clean energy. In the
United States (US), the first regulations were legislated in 1970, and Environmental Protection Agency (EPA) reported a drop
of ~40% in aerosol related air pollution in major US cities in the last half century (DeMocker, 2003). In the European Union
(EU) countries, a decrease of ~29% in aerosols has been reported since 1970, as a result of policy measures and technological
improvements (Turnock et al., 2016). In South and Central American countries, a poor emission control has been reported,
resulting in 150 million people living in urban areas with poor air quality (UNEP, 2016). Jin et al. (2016) summarized the
policies and their implementation over the last three decades for China, and as a consequence the China's anthropogenic
emissions markedly declined in the last decade (Zheng et al., 2018). India launched in 2019 the National Clean Air Programme,
with the objective to reduce in 2024 the particulate air pollution by 20-30% with respect to 2017 levels, in 122 cities (Ganguly
et al., 2020). The regulations for air pollution for many African cities are weak or non-existent (Abera et al., 2020), which will
be challenging for their future air quality, since many African cities have an unprecedented population growth. So, it is of vital



importance to understand the trends of aerosol loads and their spatial variability over those great population agglomerations, assessing that way the effectiveness of air pollution emission regulations.

Aerosol related air pollution can be quantified, in optical terms, through Aerosol Optical Depth (AOD), which is the most comprehensive variable for assessing the aerosol load of the atmospheric column. Ground-based sun-photometers have been
deployed during the last 20-25 years providing long term AOD measurements at established global/regional networks such as AERONET (Holben et al., 1998), GAW-PFR (Kazadzis et al., 2018) and SKYNET (Nakajima et al., 2020). Although, the most precise method for monitoring AOD is provided by surface sun-photometers, these measurements are scarce, lacking of full spatial and temporal coverage. On the contrary, satellite remote sensing is a powerful tool for monitoring AOD around the globe (Kaufman et al., 2002) at considerable accuracy, almost on daily basis and at relatively fine spatial resolution. Despite
that the quality of spaceborne AOD over urban surfaces depends strongly on the limitations of the retrieval algorithms (Gupta et al., 2016), yet it is an aerosol parameter worldwide available at high spatial and temporal resolution.

There is a large number of studies dealing with AOD trends at local (e.g. Raptis et al., 2020; Vohra et al., 2021), regional (e.g. Che et al., 2019; Cherian & Quaas, 2020; Zhao et al., 2017) and global scale (e.g. Buchholz et al., 2021; G. Gupta et al., 2022; Logothetis et al., 2021; Wei, Peng, Mahmood, et al., 2019). AOD trends (either from spaceborne or from ground-based
observations) in the last two decades indicate robust regional patterns, with decreasing aerosol loads over Europe and US since 2000 and reversed to decreasing since 2010 for East Asia and continuously increasing over southern Asia, which are also supported by in situ particulate concertation measurements (Gulev et al., 2021 and references therein). Nevertheless, limited studies have focused on aerosol regime over megacities, which are the major anthropogenic aerosol sources worldwide and hence studies focused on them provide the direct link between aerosol load and emissions variability. In order to conduct
studies like this, it is necessary to have available a dense surface based network or to take advantage of the satellite remote sensing coverage capabilities. Alpert et al. (2012) relying on spaceborne aerosol measurements (MODIS and MISR) investigated the AOD tendencies over the 189 largest world cities for the 8-year period between 2002-2010. According to his findings, increasing AOD trends were found for the largest cities in the Indian subcontinent, the Middle East and North China whereas opposite trends were evident in European, northeast US and southeast Asian megacities. The aim of this study is to
use up-to-date and state of the art spaceborne aerosol retrievals (AOD and Dust Optical Depth - DOD) to study local and transported aerosols that can affect air quality in large cities around the globe. Based on the idea of Alpert et al. (2012), we used a finer spatial resolution (0.1°) aerosol product, that is able to reveal aerosol air pollution disparities within megacities and will allow for a more detailed AOD spatial analysis at urban scale and a period from 2003 to 2020 in order to have more robust trends. Analyzing the aerosol spatiotemporal variability over megacities is of great importance for their air quality and
human health.

The present study aims to investigate the aerosol regime of megacities, by addressing the following scientific questions:

- What is the spatial variability of AOD in megacities at different geographical locations?
- What is the temporal variability?



- Are surface-based AOD measuring stations adequate on reflecting the spatiotemporal changes of AOD in megacity scales?

## 2 Data and methods

### 2.1 Megacities information

Megacities are defined as cities with population of more than 10 million, but in our analysis, we include cities with population between 5 and 10 million as well, due to their potential to become megacities in the coming decades as estimated by UN projections (UN, 2018a). Table 1 summarizes the number of cities according to their population by 2018 and future projections. According to UN projections, 10 cities with population between 5 and 10 million are expected to become megacities in the near future. Here, we focus on 81 cities, with more than 5 million inhabitants as reported in 2018, which are listed in Table A1 (UN, 2018a, 2019a) and their geographical location is depicted in Fig. 1.

It should be mentioned that the utilized population data, extracted from the UN database, corresponds mostly (65%) to urban agglomerations, which are the areas with city's boundaries the geographical limits of the contiguous urban area (or built-up area). A small part refers to the metropolitan areas (25%) and even smaller (10%) to the city proper. For a Major Metropolitan Area (MMA) more than one city are close together, in most of the cases with no distinct limits. For example, the Kinki MMA, with Osaka being the city with highest population, and the Los Angeles metropolitan area including Long Beach and Santa Ana population. There are 5 MMAs included in this study and they are denoted with different colour in Table A1. One last remark concerning population data is about Guangzhou, Shenzhen, Hong Kong, Dongguan and Foshan (also differently marked in Table A1). They are all located in the Guangdong – Hong Kong – Macau Greater Bay Area. This area, which is confined in a domain narrower than 1° x 1°, also includes other 5 cities with population more than 1 million forming a megalopolis which is the largest urban agglomeration on the planet. The cities in this particular region and the MMAs will be exempted from the spatial gradient analysis.

**Table 1.** Number of cities according to their population (adopted by UN (2018a)).

|  | **By 2018** | **Projection to 2030** |
|---|---|---|
| **Megacities** <br> (≥ 10 million) | 33 | 43 |
| **5-10 million** | 48 <br> (10 of those -20% - are projected to become megacities by 2030) | 66 |
| **Total ≥ 5 million** | **81** | 109 |
| **1-5 million** | 467 | 597 |





| | 548 | |
|---|---|---|
| **≥ 1 million** | 28 of those -5% - are projected to cross the 5 million mark) | 706 |
| **500,000 -1 million** | 598 | 710 |

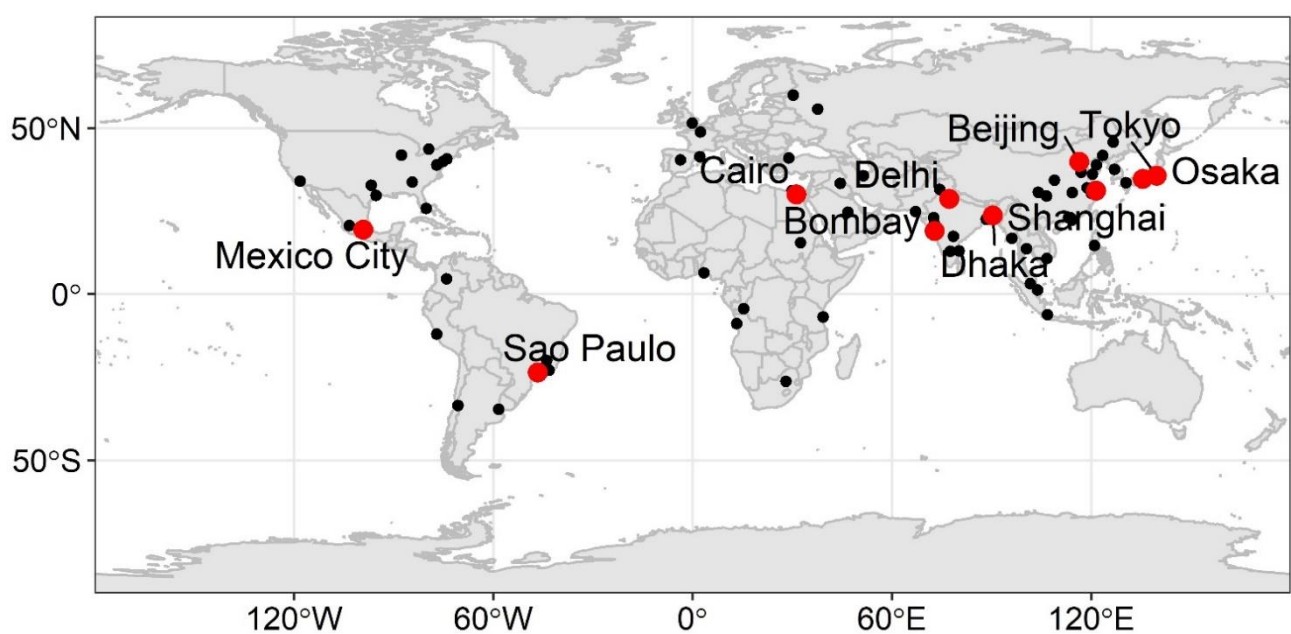

**Figure 1.** Map of the location of the 81 cities with population greater than 5 million (black circles). With red circle are denoted the 10 megacities with the highest population.

**2.2 Spaceborne Aerosol Optical Depth (AOD) and Dust Optical Depth (DOD)**

Daily retrievals of AOD at 550nm from the MODerate resolution Imaging Spectroradiometer onboard the Aqua satellite (MODIS-Aqua) were used for this analysis. Specifically, quality assured Collection 6.1 MODIS – Aqua Level 2 AOD retrievals from the Dark Target (DT) and Deep Blue (DB) were merged to one dataset and presented by Gkikas et al. (2021). Based on those MODIS AOD retrievals and on MERRA-2 (Randles et al., 2017; Gelaro et al., 2017) reanalysis data, the MIDAS (ModIs Dust AeroSol; Gkikas et al. (2021, 2022)) dataset was developed providing columnar mid-visible (550 nm)

AOD and Dust Optical Depth (DOD), on a daily basis, at fine spatial resolution (0.1°x0.1°) and at global scale over the period 2003-2020.

**2.3 Ground based measurements**

AERONET ground-based measurements of AOD for cities that have available long-term time series in the 2003-2020 period were also analysed. The Level 2, Version 3 daily mean product of AOD at 500nm was collected for stations with at least 8



years of data within the study period. Using the Ångström exponent 440-870nm, the AOD values were interpolated at 550nm. Table B1 gives the details of the 27 AERONET stations utilized for the comparison. Statistics and trends of surface-based measurements were calculated with the same methodology and data availability criteria applied to the satellite data (see Sect. 2.5).

## 2.4 Spatial features

Initially, the geographical distribution of long-term (2003-2020) mean annual and seasonal AOD was derived for a square area spanning of ±1° around the city centre (as defined at UN (2018b) database). The goal of this analysis was to investigate the AOD variability inside the urban areas, as retrieved by spaceborne data. In addition, visual inspection of aerosol distributions over 1°x1° areas and the identified patterns, was used for the first qualitative classification of megacities. The main advantage of the MIDAS AOD dataset is the high spatial resolution (of 0.1°) and the daily data availability. However, there were

challenges during averaging regarding the data availability and for this reason temporal availability criteria were applied. Every seasonal mean value was calculated when at least 9 days were available (1st threshold applied), whereas the annual means were computed when at least three seasonal means were valid (2nd threshold applied). Based on the filtered year-by-year seasonal and annual means their respective long-term averages were calculated. In Fig. 2, the geographical distributions of the annual (Fig. 2b) and seasonal (Fig. 2d) long-term averages are illustrated for the megacity of Tokyo. Comparing the AOD field ±1°

around city's centre (black circle) with the corresponding Google Earth satellite map (Fig. 2a), higher values of mean AOD values can been seen over the urban agglomeration of Tokyo, with respect to the surrounding area, throughout the year (Fig. 2b, d). The blank boxes are pixels that do not fulfill the applied criteria for data availability or there are not available data. This example case demonstrates the applied methodology as it will be explained in the following sections.





**Figure 2. (a) Geographical limits of the study area for the megacity of Tokyo on © Google Earth maps. (b) Geographical distribution of the long-term (2003-2020) annual averaged AOD for the broader area of Tokyo megacity. (c) Spatial representation equal lat/lon grid of AOD data and the sectors under investigation with different colours. (d) Geographical distribution of the seasonally averaged AOD for the broader area of Tokyo megacity. Blanc pixels do not fulfil the data availability criteria. The black circle denotes the pixel of the megacity centre.**



In order to investigate further the spatial AOD variability, we quantified the AOD gradients between city's centre and the surrounding areas. The square area at each city was divided into 6 different sectors (Fig. 2c). Sector 1 (or S1) is the 4-pixel area that encompasses the city center. The daily AOD time-series has been constructed by calculating the AOD median for

each sector under investigation. This aggregation process increased the data availability compared to the single pixel approach. Following the previous procedure with filtered seasonal and annual means, the long-term average AOD for every sector was derived. An example for Tokyo is given in Fig. 3. For this megacity a mean AOD value ~0.35 was found over the city's center and decreasing mean AOD values moving towards the outer sections. Although, a uniform approach like this ignores the effect of topography (mainly mountains and sea), that breaks the symmetry around many cities, it could be considered an indicator

of the spatial distribution of AOD. It should also be highlighted that the "city center" area could have different characteristics and be wider than the area assumed in this approach. These assumptions are used only to make the results comparable and studies focusing on a small number of cities, should make a specific division for each case. In order to give a single number that will describe the spatial gradient of AOD field from megacities center to the surrounding area, linear regression was performed to the sectoral annual AOD averages in order to calculate the AOD changes per 0.1° along with their statistical

significance. The results then are expressed in Δ(AOD) per 1° and are used for a categorization of the megacities.

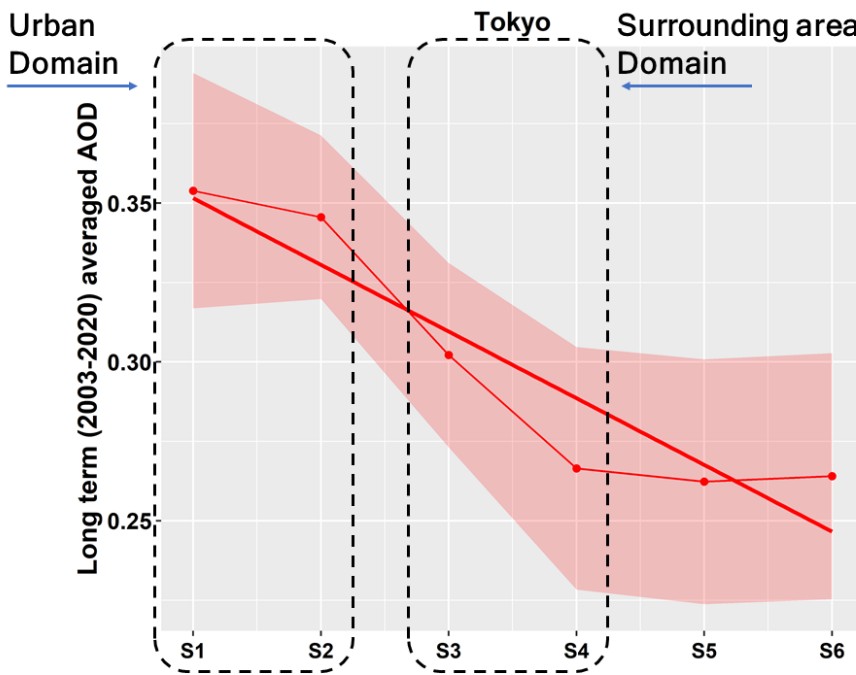

**Figure 3. The long-term average of AOD (red points) for Tokyo megacity for the 6 different sectors considered around every megacity centre. The shaded area denotes 1σ. The solid red line is the regression line resulted from the 6 points.**

Another source of uncertainty for the unified pixel-based approach that was followed for all cities would be the East-West (E-W) direction size of the pixel. More than 50% of the cities are located at absolute latitudes $25 - 45°$ (left panel Fig. 4) with a median size of 10 pixels (=1°) in E-W direction of 95km (central panel Fig. 4), and with majority of the cities (over 75%) within the ±10km limit. Even for the high latitude cities (~60°) the E-W size of the 10 pixels (=1°) is ~65km (~2/3 of the median). This difference for the high latitudes cities has been considered only in the subsequent 2-domain analysis (see Sect.

2.5). The right panel of Fig. 4 shows the distribution of cities elevation, a parameter that has been also considered in the interpretation of the results.

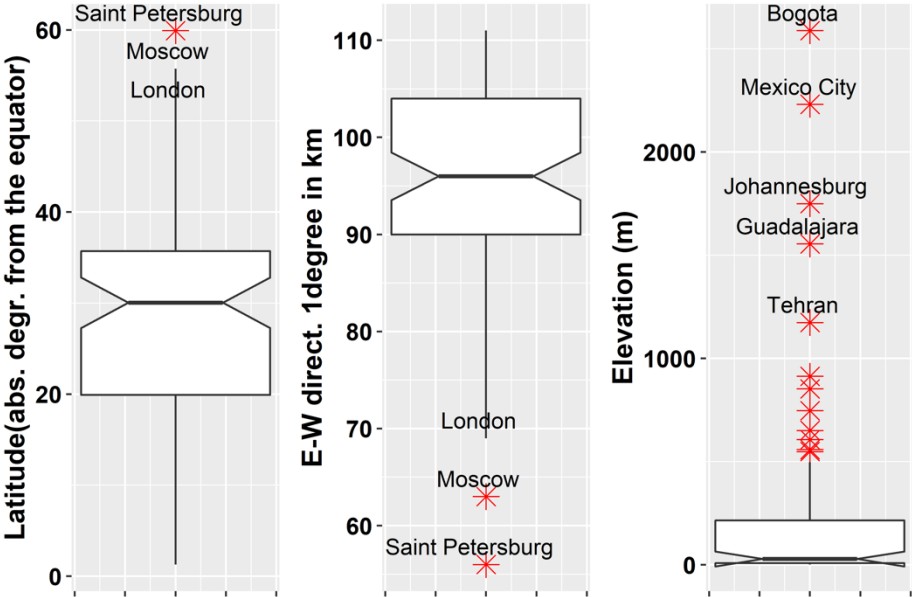

**Figure 4.** The distribution of the absolute values of megacities latitude (left panel). The distribution of the equivalent length of 1-degree East-West (E-W) direction in km (central panel). The distribution of megacities elevation in m (right panel).

## 2.5 Temporal variability and long-term averages

In order to investigate the temporal variability of AOD and derive the long-term mean values, an area of 0.4°x0.4° (4x4 pixels, S1 and S2 combined) was considered as representative of the urban agglomerations, denoted as urban domain hereafter (Fig. 5). In the same manner as in 6-sector analysis, the daily AOD time-series has been constructed for the urban domain by calculating the AOD median. Filtered (the same temporal criteria as in Sect. 2.4) annual mean AOD values were calculated

and based on them the long-term (2003-2020) annual mean values were derived for the urban domain. The AOD interannual variability and decadal trends for the urban domain were derived, and the methodological details are described in the following subsections. A 4x6 pixels area (instead of 4x4) was considered for high latitude cities (London, Moscow and Saint Petersburg), in order to have comparable areas for all cities. The differences in annual mean AOD were 1-5% compared to the same results without making this correction. Despite the small increase in data availability (~10%) when this correction was applied, the

trend calculations remained stable. In general, the uncertainties introduced due to the pixel size are minor compared to those

associated with the data availability.

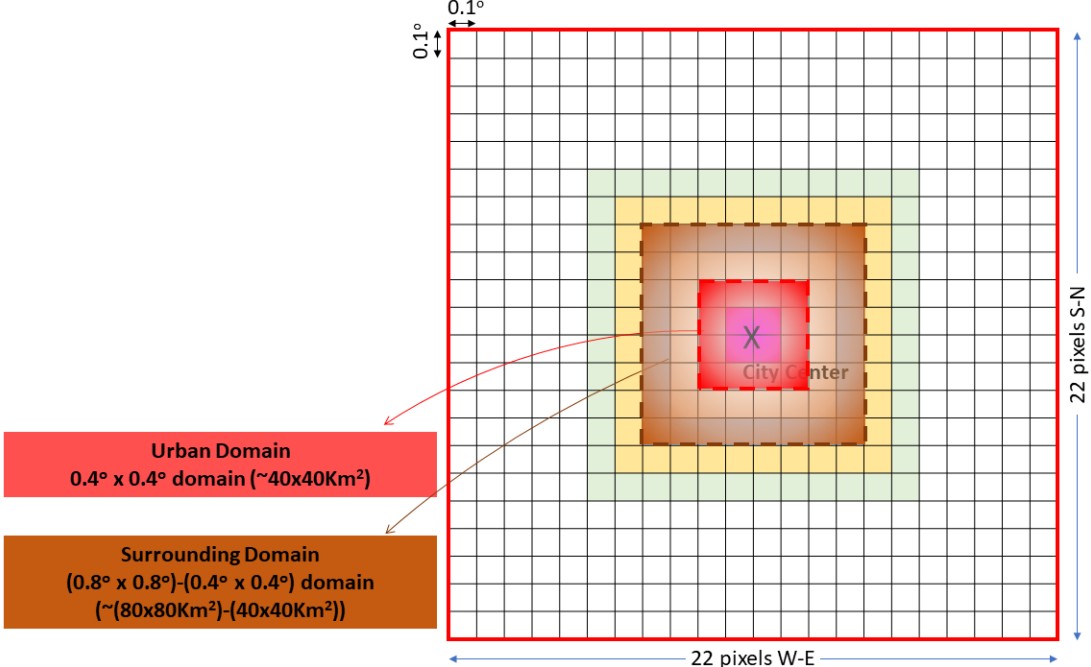

**Figure 5. Spatial representation of equal lat/lon grid of the AOD data and the urban domain (red) and surrounding area domain (brown) under investigation.**


**AOD Interannual variability and trends**

The linear trends were calculated by using simple linear regression for the filtered annual mean values, when at least 10 years (out of 18) were available. The statistical significance of the trends was assessed by performing the t-test. In Fig. 6 we are presenting an example for Tokyo. The red and brown curves correspond to the annual means for the urban and surrounding

domains, respectively, the shaded areas represent the standard deviation and the same colored lines denote the calculated linear trends. For this interannual time-series, declining AOD trends are evident both for the urban and the surrounding domains of the Tokyo megacity.





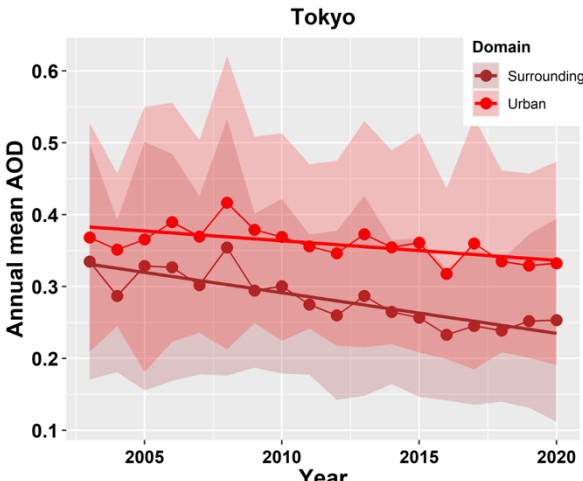

**Figure 6. Annual mean AOD values for the Tokyo megacity (red points) and the surrounding area (brown points) and the corresponding standard deviations (shaded areas). The linear trends are depicted with the straight lines.**

**AOD Intra-annual variability**

Monthly mean AOD values were calculated when 3 days were available. From those filtered monthly AODs the long-term monthly mean AOD values were derived, in order to assess the intra-annual variability of aerosol loads for every cities' urban domain. As an example, the intra-annual variability for the megacity of Tokyo is given in Fig. 7. In order to give a single measure of this intra-annual variability of AOD, the temporal coefficient of variation (CV) was derived. The mean and the standard deviation (SD) were calculated for cities with at least 9 monthly values, in order to calculate the temporal CV using the following formula:

$$CV = \frac{SD}{mean} 100\% \quad (1)$$

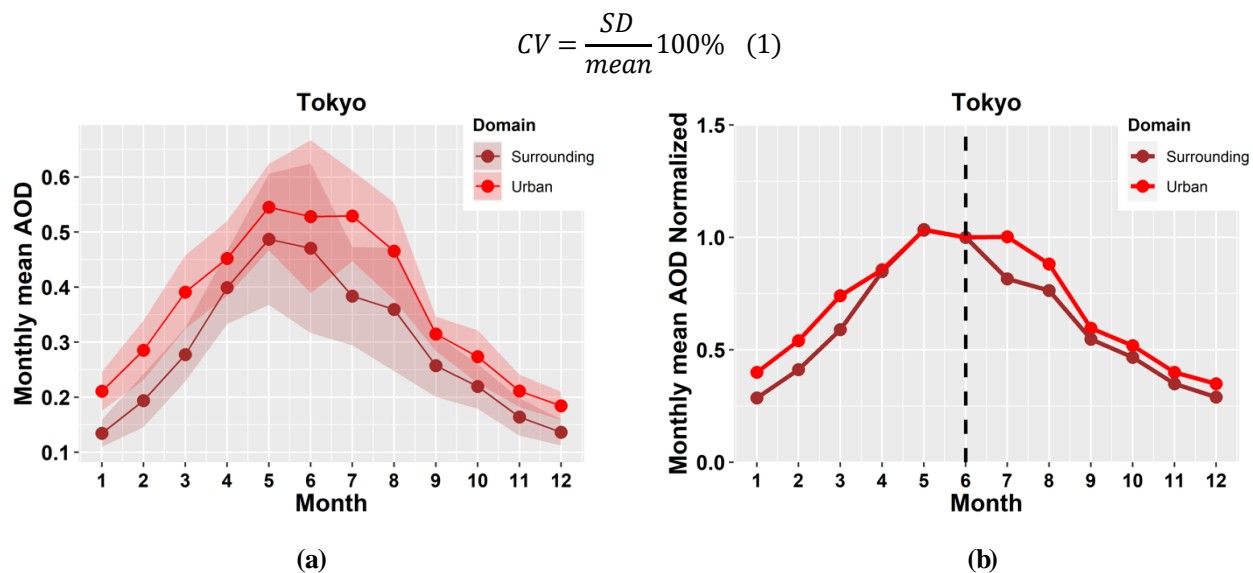

| (a) | (b) |
|---|---|

**Figure 7. (a) AOD intra-annual variability for Tokyo megacity (red line) and the surrounded area (brown line). (b) Normalized monthly mean AOD values with respect to June mean AOD value.**





### 2.6 Spatiotemporal variability

In order to investigate the spatiotemporal AOD variability, the long-term mean AOD values and the linear trends were also derived for a second domain corresponding to the surrounding area of the urban agglomerations (brownish cells in Fig. 5).

This domain (surrounding domain hereafter, Fig.5) is the remainder from a broader 8x8 pixel domain around the city's center after eliminating the urban domain (4x4 pixels). Again, for high latitude cities extra pixels were considered (12x8 pixels) adjusting for different pixel area, but even smaller differences were observed (1-4%), with respect to those found for the urban domain, considering this change.

## 3 Results and discussion

### 3.1 Megacities' spatial AOD characteristics

In this section, megacities' AOD spatial variability has been investigated and used as a proxy for cities classification.

### 3.1.1 Geographical distribution

Analysis of the mean annual and seasonal AOD geographical distributions for the 18-year period, revealed spatial AOD heterogeneities between the urban and surrounding domains for all megacities, similar to those that have been already presented
in the example for Tokyo (Fig. 2 b and d). Following the same approach for all the examined cities, some interesting features were found, which led to a first qualitative classification.

Out of the 81 cities 53% are coastal and 47% are inland. This classification is important, as coastal cities are related with low data availability -due to satellite retrieval algorithm restrictions- and most cities that are not present in the following results are from this group, because they didn't fulfil the criteria of data availability. One group consists of high-elevated (> 2000m) cities
such as the Mexico City and Bogota. Those cities are related with low data availability and for the available pixels the AOD values are very low, which is not consistent with the GB measurements (see Sect. 3.5), so they were excluded from the spatial gradient analysis. Another group contains inland cities situated nearby large deserts, where dust is regularly dominating the aerosol mixture (Cairo, Tehran, Riyadh, Baghdad and Khartoum).

Cities with few blank pixels (<50%) within the defined limits, are clustered in three distinct groups. There are cities where the
AOD fades from the central sector towards the surrounding area (~60% for coastal and ~35% for inland). This is always the case for the MMA group of megacities. For a number of cities (~40% for coastal and for inland), an opposite gradient is found attributed to the higher anthropogenic activities (e.g., industrial zones) in the outer domain. Finally, in the third category are grouped only inland cities (~25%), where a uniform distribution of mean AOD exhibiting a weak variation in spatial terms. This is a qualitative classification and it is based on the more uncertain results (based on the data at the original spatial
resolution at 0.1°). In order to give a quantitative measure of this spatial variability and with lower uncertainty compared to the singe pixel approach, the six sector and two domain analysis was performed (see Sect. 3.1.2).



### 3.1.2 Long-term means and spatial variability

One of the objectives of this analysis was to take advantage of the high spatial resolution (0.1°) of MODIS AOD product in order to assess inequities in aerosol air pollution exposure levels within those high-risk communities. The aggregation of the
AOD for the different sectors and domains that have been applied (see Sect. 2.4 and 2.5), increase the confidence in the results relative to the single pixel approach. It must be pointed out here that, the combination of the local topography and the location of the anthropogenic activity outside the examined cities (industrial zones, nearby smaller cities) distorts the spatial AOD distribution, highlighting the possible non-homogeneity of the city sectors or domain of the surrounding area we have used.

### 3.1.2.1 Spatial (six sector) analysis

In order to give a single measure of the AOD gradients (i.e., sharp or smooth) in the vicinity of the megacities, a linear fit was applied to the 6 sectoral 18-year mean AOD values. The slopes are expressed as Δ(AOD) per 1° and along with their statistical significance are presented in Fig. 8. Negative values indicate that greater values of AOD were found in the city centre and are denoted with blue colour. Positive values are related with higher AOD values moving away from cities centre and are denoted with red colour. Due to low data availability, 24 cities were discarded from the analysis. The MMAs and cities with high
elevation (>2000m) were also excluded from this spatial gradient analysis. In Appendix C for every city the mean AOD values for six sectors are given, normalized with the 18-year mean value of S1 (Fig. C1).

For the majority of megacities (~ 65%), lower mean AOD values resulted for the surrounding areas with respect to the city's centre (i.e., negative Δ(AOD)). This percentage is raised to ~75% when only the statistically significant results are considered. The largest values of negative AOD gradients were found for Xian (China), Alexandria (Egypt) and Santiago (Chile), attributed
to complex topography (high mountains and coastal areas). On the other hand, only 19 (~35%) cities have larger mean AOD values in the surrounding sectors (i.e., positive Δ(AOD)) and most of them are located in China and India. The reason that higher AODs were recorded in the surrounding sectors for most of those cities is that they are enclosing smaller-scale cities (MMAs have been already excluded from the analysis), which, however, host significant sources of anthropogenic aerosols. For example, Shanghai is the Chinese city with the largest positive AOD gradient attributed to the influence from the nearby
megacity Suzhou at the west and the Pacific Ocean at the east part of the city.



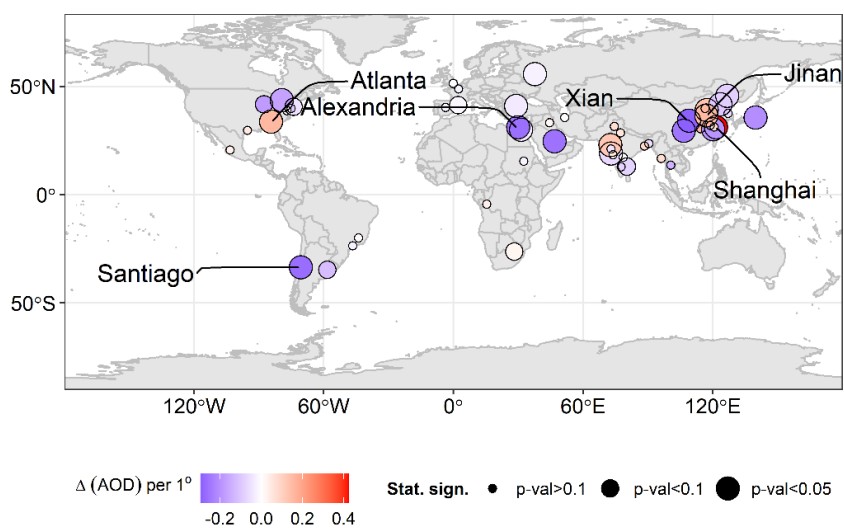

**Figure 8. Spatial AOD gradients per 1 degree (Δ(AOD) per 1°) from city center to surroundings. The circle radius is proportion statistical significance of the trends. Three cities with greatest negative AOD gradient are Xian, Alexandria and Santiago and the three cities with greatest positive AOD gradient are Shanghai, Jinan and Atlanta.**

### 3.1.2.2 Two domain spatial analysis

The geographical distribution analysis (Sect. 3.1.1) revealed that the 4x4 pixels area is a domain well fitted to the cities' urban agglomeration (urban domain). The remainder between the urban domain and an area up to 8x8 pixels (surrounding domain) was found also to be representative for the surrounding area of all cities. The results for this two-domain analysis are presented in this section.

The geographical distribution of the long-term mean AOD values for the urban domain is presented in Fig. 9. Four cities (BOGO, HCMC, KUAL, SING) are not included, due to low data availability. Low mean AOD values were found for European and American cities in contrast to Asian and African cities. Specifically, the larger mean AOD values (>0.5) were recorded for Chinese cities, with Indian cities following, since both areas are densely populated and with high industrial activity. Many megacities (mostly Asian) lying in the proximity of great deserts are also influenced by natural aerosols of desert dust (Proestakis et al., 2018; Gkikas et al., 2022). In order to investigate further this feature and quantify this influence, the long – term mean DOD and the DOD to AOD ratio – as a metric of dust contribution in the aerosol mixture - were derived using the MIDAS DOD and AOD products and the obtained results are presented in Fig. 10. Significant dust contribution (~20-40%), was found for the western Chinese cities, with mean DOD values ranging from 0.1 to 0.2, that were influenced by the nearby arid and semi-arid regions (namely the great Gobi Desert and the Taklamakan Desert). For Indian and Pakistan cities around the Thar Desert (India – Pakistan borders), even bigger dust contribution was found ~ 40-50% and mean DOD values ranging from 0.15 to 0.35. The greatest dust contribution (up to 60%) with significant mean DOD values up to ~0.4


was found for the African cities of Khartoum and Cairo and for all Middle East cities (Riyadh, Baghdad and Tehran) that are influenced by the great deserts of North Africa and Middle East.

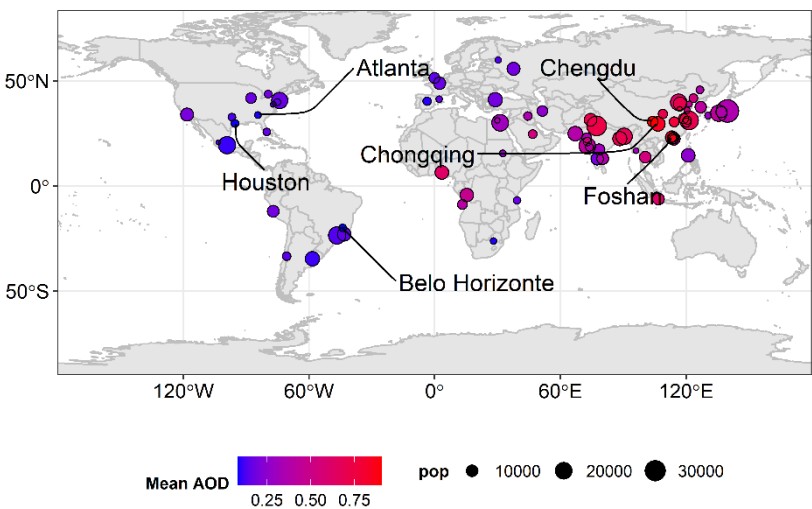

**Figure 9. Long term (2003-2020) mean AOD for megacities urban domain (~40kmx40km around cities centre). The circle radius is proportion to cities population. Three cities with greatest mean AOD are those of Chengdu, Foshan and Chongqing and the three cities with lowest mean AOD values are those of Atlanta, Houston and Belo Horizonte.**

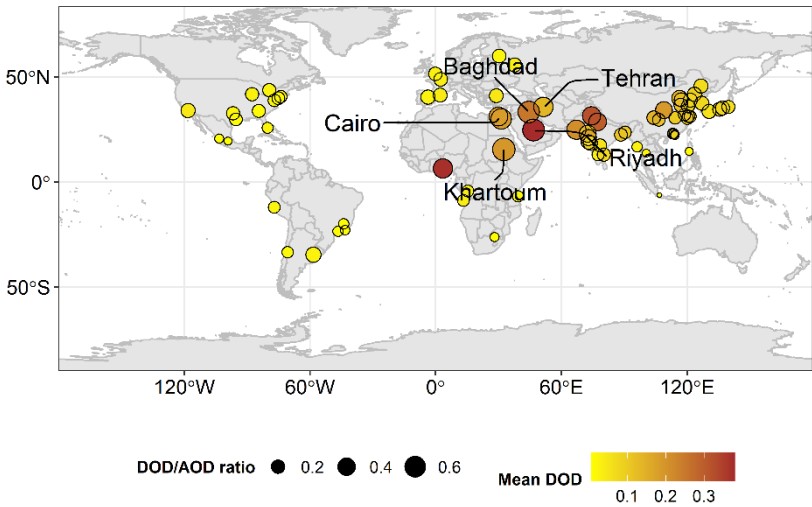

**Figure 10. Long term (2003-2020) mean DOD for megacities urban domain (~40kmx40km around cities centre). The circle radius is proportion to DOD to AOD ratio. Five cities with greatest dust contribution (mean DOD/AOD ratio) are Khartoum, Riyadh, Baghdad, Cairo and Tehran.**





In order to contrast the mean AODs for the urban domain against those of the surrounding area we have reproduced a global scatterplot with their matchups, presenting also ancillary information (i.e., continent, population, coastal/inland) (Fig. 11). Points residing on top of the one-by-one line indicate cities with homogeneous spatial AOD distributions (inside and
surrounding area), whereas above/below the equality line AODs are higher in the surrounding and urban domain, respectively. Almost 60% of the cities have greater mean AOD values over the urban domain and 40% have greater values over the surrounding domain. There are more cities with differences exceeding 10% when higher values of AOD are recorded over the urban domain compared to the opposite case. In general, the obtained results are in line with those of spatial gradients (sectoral analysis, Fig. 8), while almost all cities (apart from 4) are included here, due to the greater data availability for the two-domain
approach. Shenzhen, Osaka and Jakarta show the biggest decrease of 0.15-0.25 (~ 25-35%) in a range of few kilometres from their centre. On the other hand, Chinese cities (Shanghai, Suzhou and Qingdao) and Atlanta have greater mean AOD values in the area surrounding the centre.

Additionally, the classification of the cities according to their geographical location (in line with Fig. 9) came up, according to the obtained results for both domains. All European and American cities yield mean AODs ranging from 0.08 to 0.20, in
contrast to African and Asian cities in which the corresponding levels range from 0.25 to 0.9 (apart from Bangalore (0.2), Dar es Salaam (0.17) and Johannesburg (0.09)).

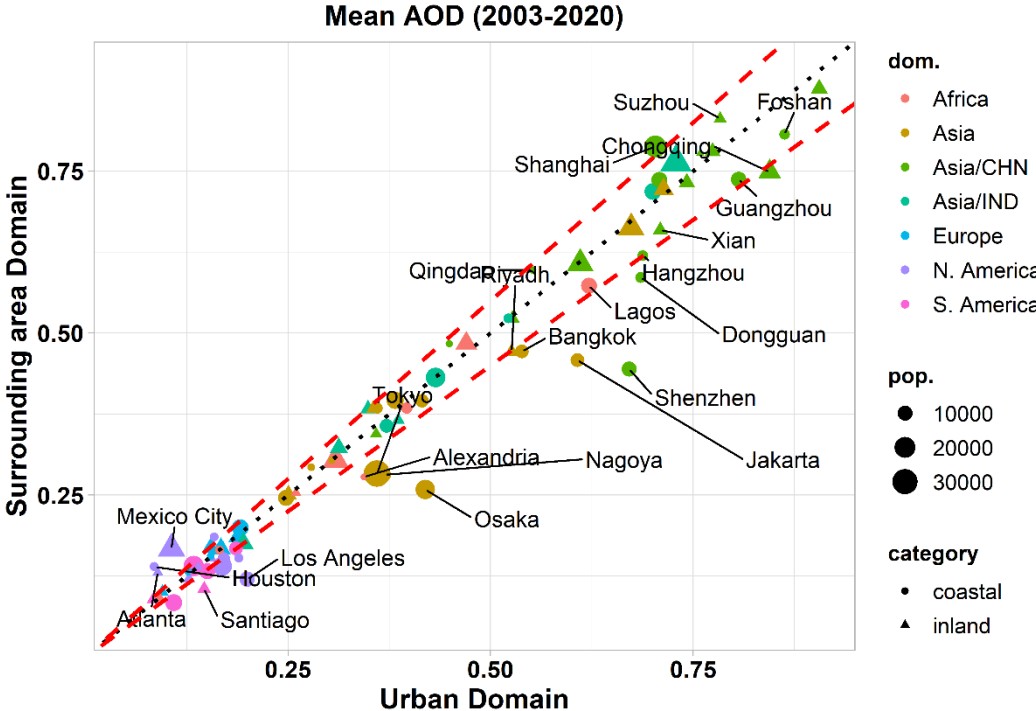

**Figure 11. Long term (2003-2020) mean AOD for megacities' urban domain (~40kmx40km around cities centre) versus the surrounding area domain. The name of the cities with difference in mean AOD greater than +/-0.04 between the two domains is**
**depicted. The dotted black line is the identity line and the dashed red lines denotes +/-10% difference in mean AOD between the two domains.**





### 3.2 Temporal variability

#### 3.2.1 AOD trends

The geographical distribution of the AOD changes per decade (Fig. 12, after excluding eleven cities which didn't fulfil the
temporal criteria) revealed pronounced regional features. The resulted AOD trends for the megacities are reflecting the changes
in the anthropogenic emissions, associated with the air quality regulations implemented throughout the years, in the same
manner that previous studies have shown the connection of the satellite observed AOD trends with the implemented mitigation
policies in regional scales (Gupta et al., 2022; Zhao et al., 2017)

For all European and US/Canadian cities decreasing AOD values were found (up to ~0.03 and ~0.07 per decade, respectively),
in accordance with the AOD decrease in the western Europe and Eastern North America that have already been reported in the
literature and was associated with a series of air quality control measures that have been implemented (Gupta et al., 2022;
Zhao et al., 2017). While negative trends have been found in this study for Los Angeles, other recent studies report small
positive trends for the western United States of America that are associated with reduced precipitation and increased fire
activity over the area (Gupta et al., 2022; Cherian and Quaas, 2020). However, the latter studies refer to much wider areas and
not to the city of Los Angeles, which indicates that the trends in the city may be related to different mechanisms (i.e., reduction
of anthropogenic emissions dominates over the increase due to increasing dust and smoke events) relative to the trends over
the wider region of the western US. Statistically significant negative AOD trends were derived for the Eastern Asian
megacities, with the highest values (in absolute terms) up to ~ 0.3 per decade being evident for the Chinese megacities. This
result (net negative AOD trend for the whole study period for Chinese cities) is in agreement with recent studies that reported
AOD decrease for eastern China (Gupta et al., 2022), which was associated with the implemented emission control policies.
Specifically, for China, while up to 2010 AOD was increasing (e.g. Hsu et al., 2012) due to the rapid economic and industrial
development of the country, after 2011 declining AOD trends have been recorded for the central and eastern sectors of the
country, related with the reduction in anthropogenic aerosol emissions due to the implementation of emission control measures
(Zhao et al., 2017; Sogacheva et al., 2018). According to Sogacheva et al., (2018) the gradual AOD decline after 2011 is more
prominent for the highly populated and industrialized southeast China regions being in agreement with our results.

On the contrary, strong positive AOD trends (ranging from 0.07 up to 0.16 per decade) were found for all megacities in the
Indian subcontinent reflecting the increased industrial and financial development during this period and is in agreement with
previous studies (e.g. Buchholz et al., 2021). In a recent study by Samset et al. (2019), they have shown that the climate
implications of this dipole pattern of positive AOD trends over southern Asia and strong negative values over eastern Asia
observed since 2010, might be strong not only on a regional scale, but also for areas away from the sources.

Positive AOD trends were also found for the Middle East megacities, ranging from 0.03 to 0.1 per decade. This finding is in
line with the AOD trends that resulted from the analyses of retrievals from different satellite sensors (Che et al., 2019), although
negative trends have been reported in the study of  (Gupta et al., 2022), who analysed AOD from CALIOP. For the Latin
American megacities, the highest AOD trends are positive (~0.03 per decade for Buenos Aires and Lima), reflecting the poor





emission control measures over this geographical domain. In recent studies (Gupta et al., 2022; Buchholz et al., 2021), a reduction in AOD was reported for South America, attributed to the decline in forest fires, which might explain the negative AOD trends for the rest Latin American cities. Sub-Saharan Africa is an area for which AOD decrease has been reported in previous studies, which is opposite to our foundlings (+0.06 per decade for Luanda and +0.03 per decade for Dar es Salaam). For this area, an increase in AOD was also found by the recent study of Gupta et al. (2022), which was associated with the

increase in biomass burning of agricultural activity in the dry season over the area, but since our study is focused on megacities, our findings may reflect the increasing urbanization in combination with the limited air quality regulations over the area, but further investigation is needed here.

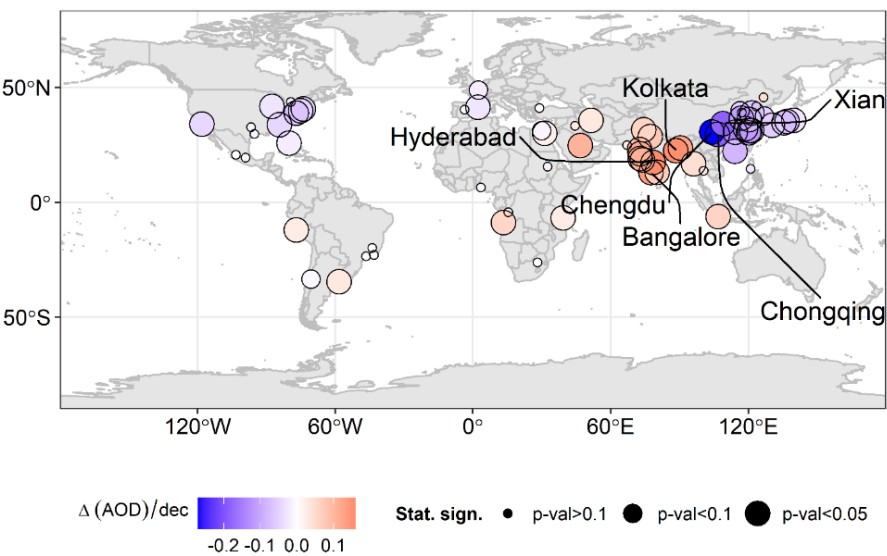

**Figure 12. Linear trend of AOD per decade for urban domain (~40kmx40km around cities centre). The circle radius is proportion**
**statistical significance of the trends. Three cities with greatest increase of AOD are Hyderabad, Kolkata and Bangalore and the three cities with greatest decrease of AOD are Chengdu, Chongqing and Xian.**

The comparison of the AOD trends against the long term mean AOD of the urban domain revealed an interesting clustering of cities, according to their geographical location (Fig. 13). It has to be pointed out here that only the statistically significant trends were included in Fig. 13, thus ensuring robust and meaningful results.

In all European and North American cities low AOD climatological values and decreasing trends were found. Note that in some US cities like Atlanta, Washington, Philadelphia and New York, apart from their relatively low mean AOD values (<0.17), considerable negative trends (ranging from -53% to -28%) were recorded.

All Indian cities result with positive AOD trends regardless of their mean AOD levels, which span from low to high values. Among them, Kolkata is the city where extremely high mean AODs (0.70) and large positive trends (+22% per decade, and





the higher in absolute values +~0.16 per decade) were revealed. Of particular interest is Bangalore, which has relatively low mean AOD value (0.20) but the maximum positive AOD trend almost +69% per decade. Bangalore's population increased from 6 million to 12 million during the last two decades, that is one of the biggest population increases for cities at this scale and thus it can explain at a large degree the extreme increase of AOD. Meanwhile, financial development in the area is linked more with new technologies and not heavy industry, which partially answers the relatively low mean.

All Chinese cities are subjected to AOD decrease (up to ~30%) while at the same time they have the highest mean AOD values ranging from 0.45 to 0.91. Chengdu is the city with the biggest mean AOD value along with the highest AOD decrease of ~30% per decade (or ~0.3 per decade).

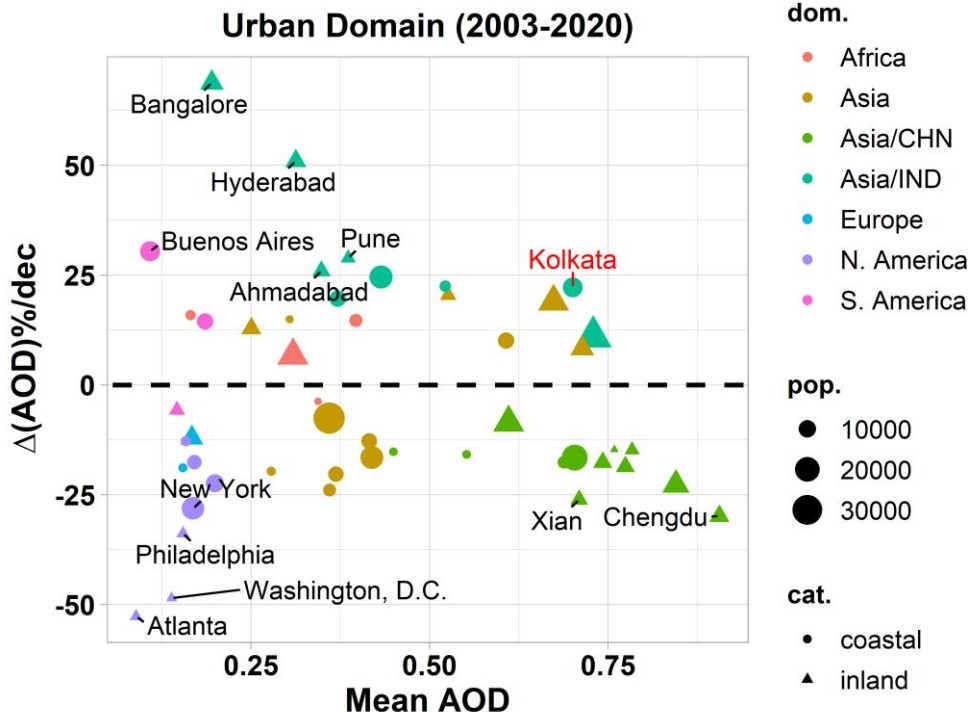

**Figure 13. AOD decadal changes expressed in percentages (Δ(AOD)% per decade) versus the long-term mean AOD for the urban**
**domain. Only statistically significant AOD trends are presented (P<0.1). Cities with absolute values of Δ(AOD)% per decade greater than 25% are denoted in the figure.**

### 3.2.2 Intra-annual variability

AOD exhibits strong intra-annual variability (Zhao et al., 2018), which is quite important to be analysed at megacities' scales in order to improve our understanding regarding the aerosol related health effects. The normalized (using as reference the mean
AOD of June) monthly mean values for all cities are provided in the Appendix D, Fig. D1, classified for the different geographical domains. In order to investigate the intra-annual variability of AOD at city level, the temporal CV (Sect. 2.5) was calculated. The geographical distribution of temporal CV is illustrated in Fig. 14, only for cities complying with the defined temporal criteria (four cities were omitted). Low CV values (<20%) were found for the eastern India, most of the Chinese





cities and for the eastern Mediterranean as well. The highest CV values >70% were recorded for three US cities. In order to
investigate further those results, the relationship between temporal CV and the long-term mean AOD values for the urban
domain was examined (Fig. 15).

A non-linear relationship was found, with the gradual decrease of the mean AOD to be related with increasing CV levels. The
CV values can reach at their maximum levels (>50%) only in the group of cities with low mean AOD values (<0.25, with only
exception Luanda), and all are American cities at the East coasts. For North American cities, the monthly mean AOD values
during the boreal winter are decreasing close to zero, that resulted to this high intra-annual variability (Fig. D1b), while for the
southern American cities rather stable values were found throughout the year (Fig. D1c). For moderate AOD levels (0.25-
0.50), a limit of 50% was found for the CV values. For the cities with high aerosol loads (>0.50), which are mostly Chinese
cities, the CV values are confined to the limit 10-40%.

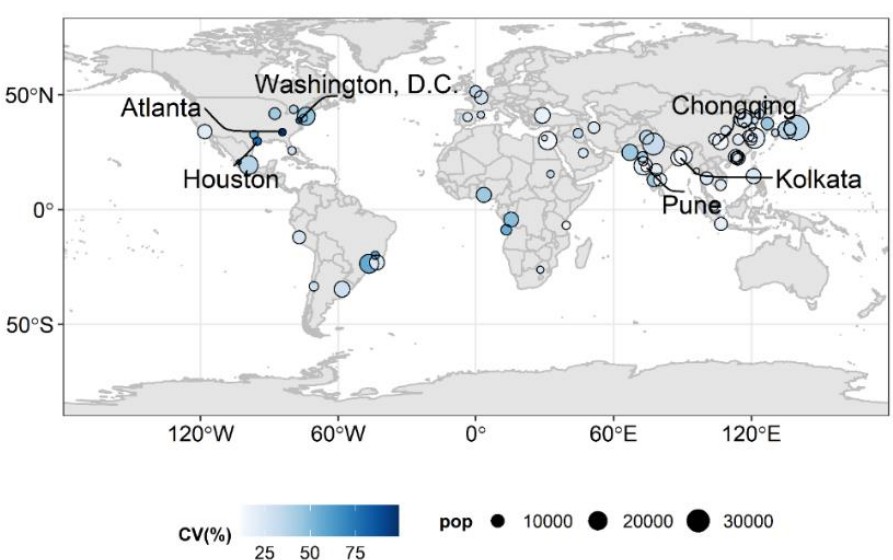

**Figure 14. Temporal coefficient of variation (CV) of monthly mean AOD (intra-annual variability) expressed in percentages for
megacities urban domain (~40kmx40km around cities centre). The circle radius is proportional to cities population. Three cities with
the highest temporal CV are Atlanta, Houston and Washington, D.C. and the three cities with lowest values are Kolkata, Chongqing
and Pune.**





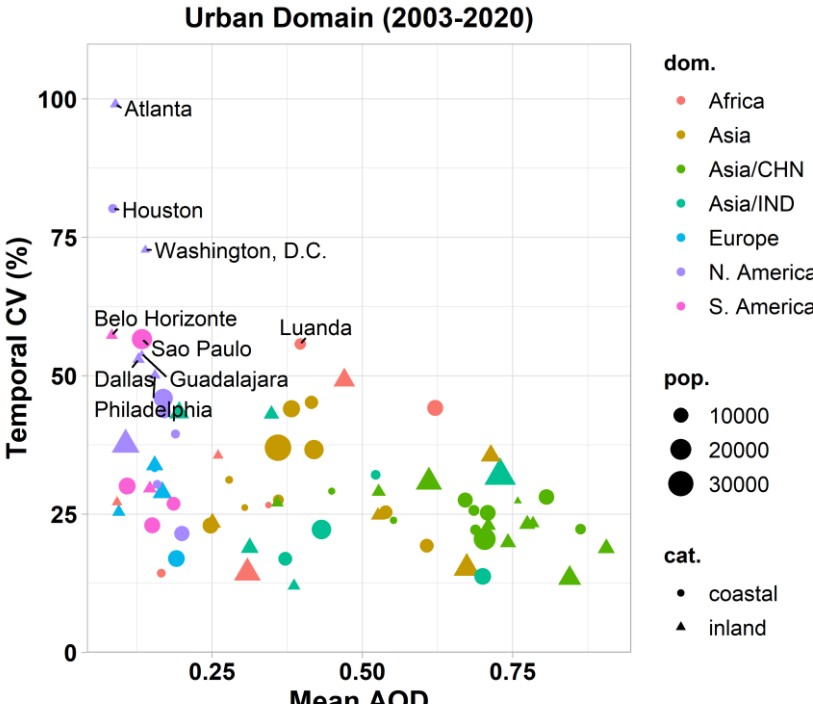

**Figure 15. Temporal coefficient of variation expressed in percentages (CV%) against the long-term mean AOD for the urban domain. The size and the colour of the points denote cities' population and geographical domain, respectively. The shape of the points denotes costal or inland cities.**

### 3.3 Spatiotemporal variability

Towards assessing the AOD spatiotemporal variability, the decadal AOD linear trends for the urban and the surrounding domains were compared (Fig. 16). We are presenting only the points in the scatterplot when the AOD trends are statistically significant (p<0.1) both for the surrounding and urban domains (43 pairs). Points residing above/below the identity line correspond to AOD trend (negative or positive) that is higher in the surrounding/urban domain, respectively.

All cities in India exhibit an increase of AOD both in the urban and surrounding domains. The largest difference was found in Hyderabad, where AOD trend is 10% larger for the urban domain, indicating an increase of the anthropogenic activity within the boundaries of the urban agglomeration. For cities where AOD decreases was found, two groups are shaped. The first one, that has greater AOD decrease for the urban domain, consists mostly of East Asian cities. Chongqing is the Chinese megacity with the largest difference (~20%), reflecting the adaptation of effective measures towards reducing city's aerosols air pollution (Gao et al., 2021). The second group, which consists mostly of the North American cities, has greater AOD decrease for the surrounding domain.





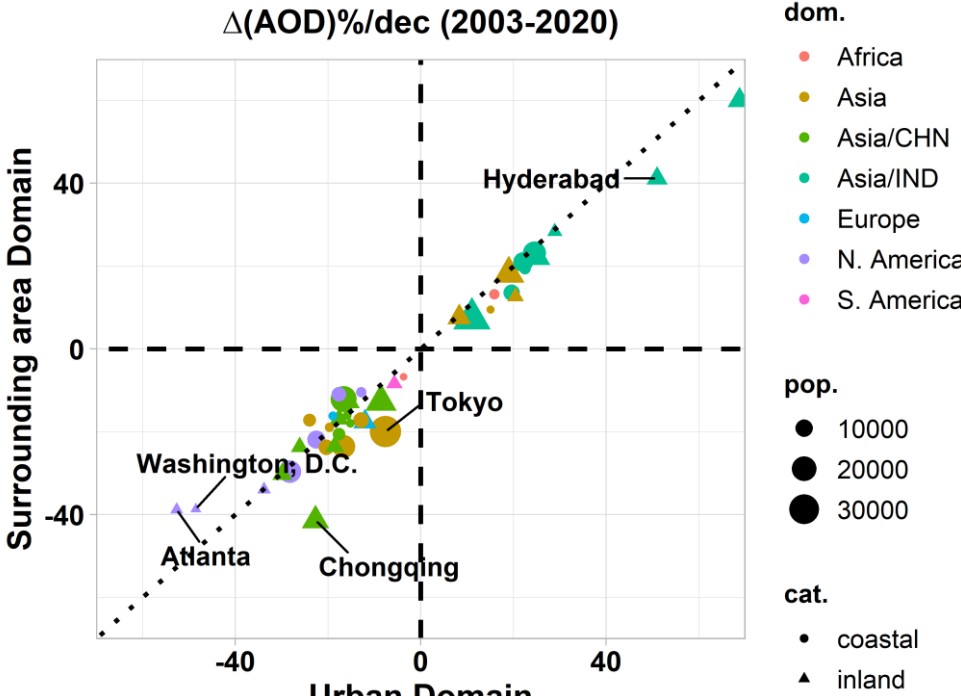

**Figure 16. AOD decadal changes expressed in percentages (Δ(AOD)% per decade) for the surrounding domain versus the urban domain. Only statistically significant AOD trends are presented (P<0.1). The dotted line is the identity line. Cities with absolute differences of Δ(AOD)% per decade greater than 10% between the two domains are denoted in the figure.**

### 3.4 Evaluation with Ground Based (GB) measurements (AERONET)

For cities with available long-term time series of AOD from GB stations of the AERONET network, an evaluation of the satellite AOD averages and trends was performed. Table B1 summarizes the information of the AERONET stations utilized. This comparison is separated to the GB stations located within the 4x4 pixel area of the urban domain and those residing in the surrounding domain of a city. The daily averaged AOD product was used from the AERONET stations to derive long-term mean AOD and linear trends, using the same approach as followed with the satellite data. The diurnal AOD variability of the AERONET data wasn't considered and this may slightly affect the long term mean AOD (Smirnov et al., 2002), but it plays a minor role for the calculation of trends.

Regarding the long-term mean AOD (Fig. 17), for the urban domain there is a good agreement between the satellite derived and the GB values (correlation coefficient R ~ 0.93). The largest deviations (expressed in percentage terms) between spaceborne and ground-based AODs are recorded in general for weak-load cities, and those deviations are maximized in the high-altitude city (>2000m) of Mexico and OSAK (limited satellite data availability). For the surrounding domain the studied cases are few, so the R is not representative. The satellite derived mean AOD is overestimated, but in this case the area that was utilized to derive the satellite results is considerable large and it is unlikely that the point measurement statistics would coincide with satellite statistics.





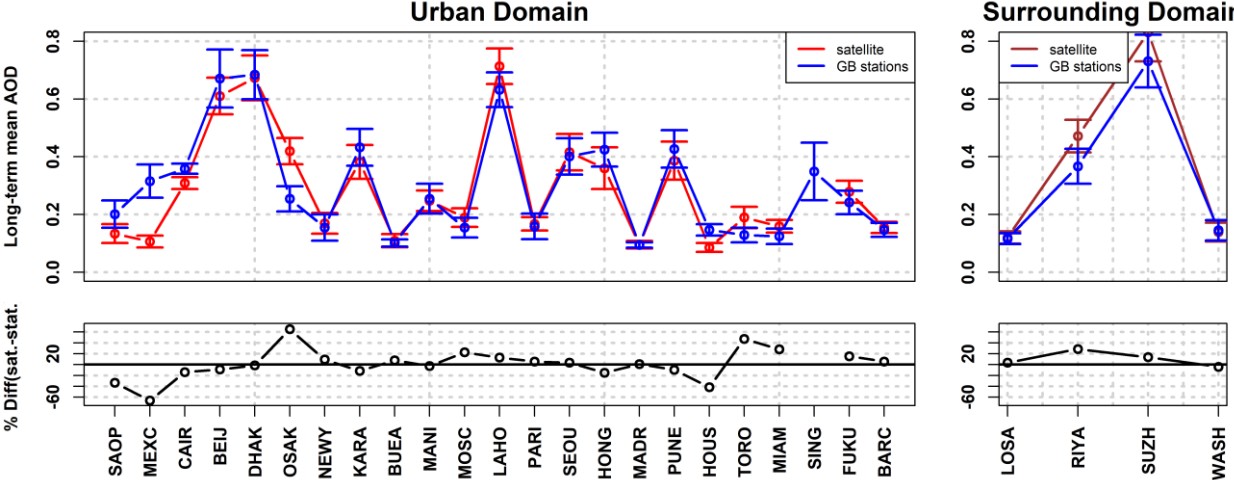

**Figure 17. Upper panels: Satellite derived (red line) and GB (blue) long term mean AOD for urban domain (left) and surrounding domain (right). Lower panel: Relative difference of mean AOD between sattellite and GB measuremnts expressed in percentage for urban domain (left) and surrounding domain (right).**

A relatively good agreement was also found for the decadal linear trends (Fig. 18), with R~0.79 for the urban domain. For 75% of cases, GB and satellite derived trends have the same sign, for both domains. In Fig. 18 are presented the results only when both spaceborne and ground-based AOD trends are statistically significant. For the urban domain, all the statistically significant trends have the same sign. For the surrounding domain, the trends' signs differ only for LOSA. This difference can be attributed to the shorter time period of measurements at the LOSA AERONET station (8 years). Additionally, in a recent study (Wei et al., 2019b), western North America was found to be the area with fewer sites with the same signs between MODIS and AERONET derived AOD trends. Nevertheless, our analysis does not focus in a validation of satellite AOD against AERONET and thus not an exact colocation of the AERONET stations with satellite pixels was attained. In order to investigate further the obtained differences, the spatial CV was calculated for the urban and surrounding domain, as the ratio of standard deviation and mean values of the pixel long-term mean AODs derived in geographical domain analysis (see Sect. 2.4). The results are presented in the lower panel of Fig. 18 along with the percentage of the available pixels inside the domain under investigation. For LOSA, the greatest value of spatial CV was found (>50%), which was anticipated, since the spatial extend of Los Angeles metropolitan area is high (including Long Beach and Santa Ana), highlighting the importance of the selection of the GB location.

An additional interesting feature is that the absolute values of all satellite derived statistically significant trends (Fig. 18) have lower magnitude compared to the GB results. This difference in magnitude could be attributed to the domain aggregated satellite values compared to point derived results of the GB stations, which may smooth out the AOD fields. In a recent study by Logothetis et al. (2021), a sensitivity analysis was performed between coarse (1°) and fine (0.1°) spatial resolution of spaceborne AOD retrievals which revealed AOD trends of lower magnitude for the coarse-spatial-resolution data. Moreover,





the daily satellite value, corresponding to satellite overpass, contains less information compared to the continuous monitoring during daytime of cloudless days of the ground-based photometers.

Hence, GB stations are representing the close area around their location and the representativeness of AERONET stations to characterize aerosol load/trends in megacities should be considered individually in each case, according to local conditions.

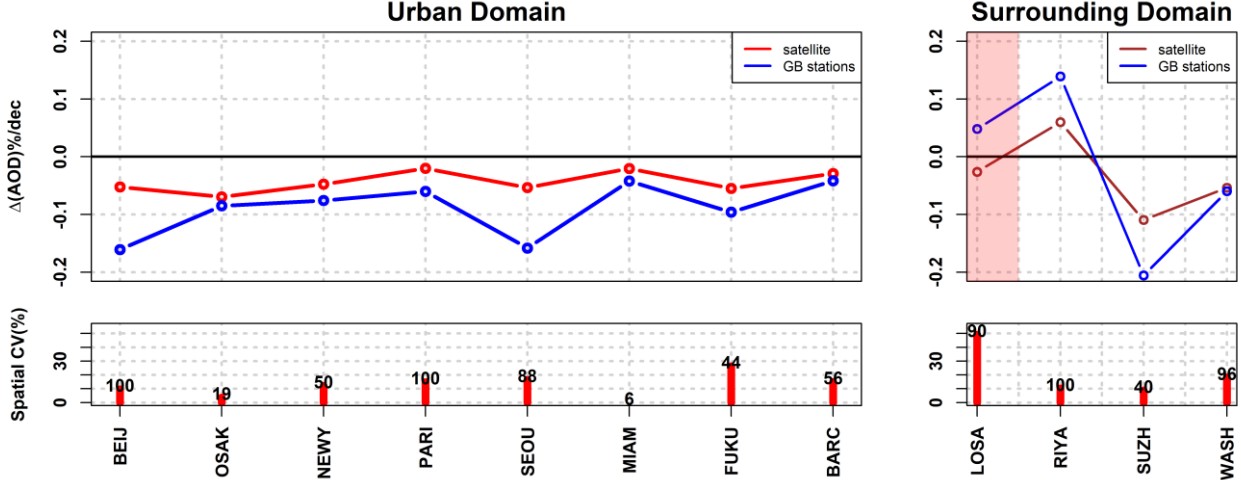

**Figure 18. Upper panels: Satellite derived (red line) and ground based (GB) (blue) statistically significant (P<0.1) AOD linear trends for urban domain (left) and surrounding domain (right). Lower panel: Spatial CV for every city and the numbers on top of the bars**
**declare the pixel availability for the domain of interest urban (left) and surrounding (right).**

## 4 Summary and conclusions

Motivated by the environmental challenges caused by the increasing urbanization and towards maximizing the use of space-borne aerosol products, in this study, we investigate the aerosol regime over the megacities of the world using satellite aerosol retrievals. We are taking advantage of the global coverage, the high sampling frequency (daily values) and the relatively fine
spatial resolution (0.1°x0.1° grid) of the eighteen years MODIS based AOD and DOD at 550nm products, in order to examine the spatiotemporal variability of aerosol loads for the largest 81 cities of the world.

For all European and American cities, mean AOD ranges mainly from 0.08 to 0.20. For all African and Asian cities but three (Bangalore (0.2), Dar es Salaam (0.17) and Johannesburg (0.09)), mean AOD ranges from 0.25 up to almost 0.90. There are cities which lie in the proximity of deserts or in the path of transported mineral dust particles, which were found to have
considerable dust contribution (up to 70%) and the associated mean DOD values (up to 0.4) can be considered as a constant background aerosol source for those megacities. Results of this contribution can be used by policy makers for defining the legislations on air quality urban thresholds.

The majority of cities (~60%) have greater mean AOD values over their urban agglomeration domain. Mostly Chinese and Indian megacities tend to have higher AOD in the surrounding areas of the city centre. Shanghai is the city with the largest
difference (13% greater mean AOD values for the surrounding domain), but in general for the cities grouped in this category



the declinations between the two domains are lower compared to the first category, because of the high mean AOD values of the urban domain. Finding inequities in the exposure at urban scales using satellite remote sensing, may be a useful tool for air pollution assessment and finally taking diverse reduction measures at community level. As AOD differences are also observed for different months/seasons during the year, AOD intra-annual variability at megacities' scale have been quantified.

Low intra-annual variability (temporal CV 10-40%) was found for Chinese, eastern Indian and Eastern Mediterranean megacities, while high values (>50%) of temporal CV was recorded for eastern American megacities.

Although Chinese cities were found with the highest mean AOD values (up to ~0.90), they also exhibit the highest AOD decrease in absolute values up to ~0.3 per decade (or 30% per decade), in response to the rigorous emission control measures implemented in the country, especially after 2010. The effectiveness of those measures also reflects the fact that, for Chinese

cities the AOD decrease was found to be higher for the urban domain (up to ~20%). Decreasing AOD values were also derived for US/Canadian and European megacities (up to ~0.07 and ~0.03 per decade) due to a series of air pollution control policies in the last decades. The maximum worldwide negative AOD trends in relative terms ~30-55% per decade were derived for most US cities, which simultaneously have low mean AOD values (less than ~0.15). The highest AOD increase worldwide in absolute (up to ~0.16 per decade for Kolkata) and relative terms (~+70% per decade for Bangalore) was found for Indian

megacities. Statistically significant positive AOD trends were found for all Indian cities reflecting the increasing urbanization and industrialization of the country. The AOD increase for Indian megacities was found to be greater (up to ~10%) for the urban domain. Statistically significant AOD increase was also found for Middle East, South African and some Latin America megacities (up to ~ 0.1, 0.06 and 0.03 respectively).

For cities where long-term ground-based AERONET measurements of AOD were available, the extent at which those

measurements can capture the spatial and temporal AOD variability, was investigated with respect to the spatial and temporal variability that were derived from satellite data. For GB stations within the urban agglomerations, a good agreement of the long term mean AOD was found (R~0.93) and with coincident sign on AOD trends in 75% of the selected stations. The resulted discrepancies are attributed, apart from the satellite retrievals related limitations (one overpass per day, high elevated pixels, low data availability etc.) and the GB retrieval limitations (large temporal gaps due to instrument issues/calibration, shorter

operating time periods, etc.) to the point (GB) versus area averaged (satellite) comparison. It was found that for areas with non-homogeneous aerosol fields (great spatial CV, e.g. LOSA in our analysis) great differences (opposite signs) were recorded between satellite and GB trends. Those findings highlight the importance of the GB station location selection for future planning of aerosol measuring sites, towards achieving representative AOD measurements for a specific city.

We acknowledge that the total column optical property of AOD, analysed in this study to describe aerosol load variability, is

not always proportional to the surface Particulate Matter (PM) concentrations, which is a parameter describing better the cities air quality, that is monitored and regulated by the cities' authorities and is related directly to health effects. However, PM concentrations are derived by in situ measurements and a dense network of those measurements is needed to describe the PM distribution of a city. Although these ground air quality monitoring stations are valuable, there are relatively few and unevenly distributed networks within a city, around the world, especially in developing countries. One way to provide consistent PM





data worldwide is by combining available PM measurements with satellite AOD observations and chemical transport models (e.g. HEI, 2020). Therefore, the use of satellite AOD is a very fitting source of information in order to have global coverage and high spatiotemporal resolution of aerosol loads over urban areas, keeping in mind that the agreement between satellite columnar AOD and ground-based PM concentrations is determined at a large degree by the vertical structure of aerosol layers (e.g. Gkikas et al., 2016).

According to our findings long term, high resolution space-borne AOD retrievals can be utilized for detecting spatial and temporal aerosol variability at an urban scale, helping towards the current and future assessments of aerosol-related impacts in megacities. The high resolution of MODIS aerosol product provides access to cities' aerosol monitoring information which can serve as the basis for health-related and other prediction services. Future work linking the AOD changes with population and emission trends in these cities will reveal the linkage and will enhance the performance of the air quality projections for

the next decades.

Appendix A.

**Table A1 Analytical table of 81 cities with highest population up to 2018 (adopted by** UN (2018a, 2019a)). **The abreviations of the statistical concept stands for City Proper (CP), Urban Agglomiration (UA) and Metropolitan Area (MA).**

| Urban Agglomerations | Short name | Country or area | 2018 Population (thousands) | Statistical concept |
|---|---|---|---|---|
| Tokyo | TOKY | Japan | 37468 | MA |
| Delhi | DELH | India | 28514 | MA |
| Shanghai | SHAN | China | 25582 | CP |
| Sao Paulo | SAOP | Brazil | 21650 | MA |
| Mexico City | MEXC | Mexico | 21581 | MA |
| Cairo | CAIR | Egypt | 20076 | MA |
| Bombay | BOMB | India | 19980 | MA |
| Beijing | BEIJ | China | 19618 | UA |
| Dhaka | DHAK | Bangladesh | 19578 | MA |
| Kinki M.M.A. (Osaka) | OSAK | Japan | 19281 | MA |
| New York | NEWY | United States of America | 18819 | UA |
| Karachi | KARA | Pakistan | 15400 | UA |
| Buenos Aires | BUEA | Argentina | 14967 | UA |
| Chongqing | CHON | China | 14838 | UA |
| Istanbul | ISTA | Turkey | 14751 | UA |
| Kolkata | KOLK | India | 14681 | MA |
| Manila | MANI | Philippines | 13482 | MA |
| Lagos | LAGO | Nigeria | 13463 | UA |
| Rio de Janeiro | RIOD | Brazil | 13293 | MA |
| Tianjin | TIAN | China | 13215 | UA |
| Kinshasa | KINS | Democratic Republic of the Congo | 13171 | UA |
| **Guangzhou**, Guangdong | GUAN | China | 12638 | UA |



| | | | | |
|---|---|---|---|---|
| **Los Angeles**-Long Beach-Santa Ana | LOSA | United States of America | 12458 | UA |
| Moscow | MOSC | Russian Federation | 12410 | CP |
| Shenzhen | SHNZ | China | 11908 | UA |
| Lahore | LAHO | Pakistan | 11738 | UA |
| Bangalore | BALO | India | 11440 | UA |
| Paris | PARI | France | 10901 | UA |
| Bogota | BOGO | Colombia | 10574 | UA |
| Jakarta | JAKA | Indonesia | 10517 | MA |
| Chennai | CHNA | India | 10456 | UA |
| Lima | LIMA | Peru | 10391 | MA |
| Bangkok | BANG | Thailand | 10156 | UA |
| Seoul | SEOU | Republic of Korea | 9963 | UA |
| Chukyo M.M.A. (**Nagoya**) | NAGO | Japan | 9507 | MA |
| Hyderabad | HYDE | India | 9482 | UA |
| London | LOND | United Kingdom | 9046 | UA |
| Tehran | TEHR | Iran (Islamic Republic of) | 8896 | CP |
| Chicago | CHIC | United States of America | 8864 | UA |
| Chengdu | CHGD | China | 8813 | UA |
| Nanjing, Jiangsu | NANJ | China | 8245 | UA |
| Wuhan | WUHA | China | 8176 | UA |
| Ho Chi Minh City | HCMC | Viet Nam | 8145 | UA |
| Luanda | LUAN | Angola | 7774 | UA |
| Ahmadabad | AHMA | India | 7681 | UA |
| Kuala Lumpur | KUAL | Malaysia | 7564 | MA |
| Xian, Shaanxi | XIAN | China | 7444 | UA |
| Hong Kong | HONG | China, Hong Kong SAR | 7429 | UA |
| Dongguan | DONG | China | 7360 | UA |
| Hangzhou | HANG | China | 7236 | UA |
| Foshan | FOSH | China | 7196 | UA |
| Shenyang | SHYA | China | 6921 | UA |
| Riyadh | RIYA | Saudi Arabia | 6907 | CP |
| Baghdad | BAGH | Iraq | 6812 | MA |
| Santiago | SANT | Chile | 6680 | UA |
| Surat | SURA | India | 6564 | UA |
| Madrid | MADR | Spain | 6497 | CP |
| Suzhou, Jiangsu | SUZH | China | 6339 | UA |
| Pune | PUNE | India | 6276 | UA |
| Haerbin | HAER | China | 6115 | UA |
| Houston | HOUS | United States of America | 6115 | UA |
| **Dallas**-Fort Worth | DALL | United States of America | 6099 | UA |
| Toronto | TORO | Canada | 6082 | MA |
| Dar es Salaam | DARE | United Republic of Tanzania | 6048 | UA |



| | | | | |
|---|---|---|---|---|
| Miami | MIAM | United States of America | 6036 | UA |
| Belo Horizonte | BELO | Brazil | 5972 | MA |
| Singapore | SING | Singapore | 5792 | UA |
| Philadelphia | PHIL | United States of America | 5695 | UA |
| Atlanta | ATLA | United States of America | 5572 | UA |
| Kitakyushu-**Fukuoka** M.M.A. | FUKU | Japan | 5551 | MA |
| Khartoum | KHAR | Sudan | 5534 | UA |
| Barcelona | BARC | Spain | 5494 | CP |
| Johannesburg | JOHA | South Africa | 5486 | UA |
| Saint Petersburg | STPE | Russian Federation | 5383 | CP |
| Qingdao | QING | China | 5381 | UA |
| Dalian | DALI | China | 5300 | UA |
| Washington, D.C. | WASH | United States of America | 5207 | UA |
| Yangon | YANG | Myanmar | 5157 | UA |
| Alexandria | ALEX | Egypt | 5086 | CP |
| Jinan, Shandong | JINA | China | 5052 | UA |
| Guadalajara | GUAD | Mexico | 5023 | MA |


Appendix B.

**Table B1. AERONET stations that have been used in the analysis and the corresponding urban agglomerations.**

| AERONET station | Period | Urban Agglomerations | Short name | Country or area |
|---|---|---|---|---|
| Sao_Paulo | 2003-2019 | Sao Paulo | SAOP | Brazil |
| Mexico_City | 2003-2018 | Mexico City | MEXC | Mexico |
| Cairo_EMA_2 | 2010-2019 | Cairo | CAIR | Egypt |
| Beijing | 2003-2019 | Beijing | CHN | China |
| Dhaka_University | 2012-2020 | Dhaka | BGD | Bangladesh |
| Osaka | 2004-2020 | Osaka | JPN | Japan |
| CCNY | 2003-2020 | New York | USA | United States of America |
| Karachi | 2006-2020 | Karachi | PAK | Pakistan |
| CEILAP-BA | 2003-2019 | Buenos Aires | ARG | Argentina |
| Manila_Observatory | 2009-2020 | Manila | MANI | Philippines |
| Santa_Monica_Colg | 2013-2020 | Los Angeles | LOSA | United States of America |
| Moscow_MSU_MO | 2003-2020 | Moscow | MOSC | Russian Federation |
| Lahore | 2007-2020 | Lahore | LAHO | Pakistan |
| Paris | 2005-2020 | Paris | PARI | France |
| Yonsei_University | 2011-2020 | Seoul | SEOU | Republic of Korea |
| Hong_Kong_PolyU | 2006-2020 | Hong Kong | HONG | China, Hong Kong SAR |
| Solar_Village | 2003-2013 | Riyadh | RIYA | Saudi Arabia |
| Madrid | 2012-2020 | Madrid | MADR | Spain |
| Taihu | 2005-2016 | Suzhou | SUZH | China |
| Pune | 2005-2019 | Pune | PUNE | India |



| Univ_of_Houston | 2006-2020 | Houston | HOUS | United States of America |
|---|---|---|---|---|
| Toronto | 2004-2020 | Toronto | TORO | Canada |
| Key_Biscayne | 2007-2018 | Miami | MIAM | United States of America |
| Singapore | 2007-2020 | Singapore | SING | Singapore |
| Fukuoka | 2012-2020 | Fukuoka | FUKU | Japan |
| Barcelona | 2005-2020 | Barcelona | BARC | Spain |
| GSFC | 2003-2020 | Washington, D.C. | WASH | United States of America |

AppendixC.

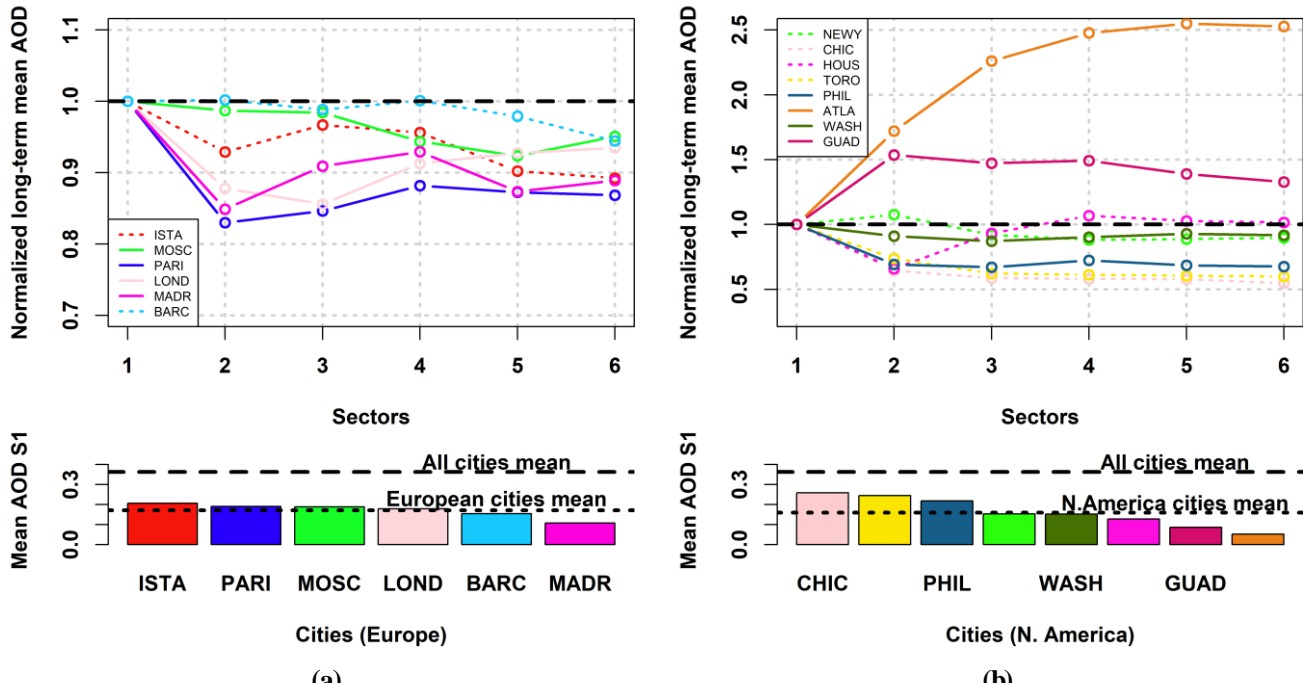







**(c)**

**(d)**

**(e)**

**(f)**





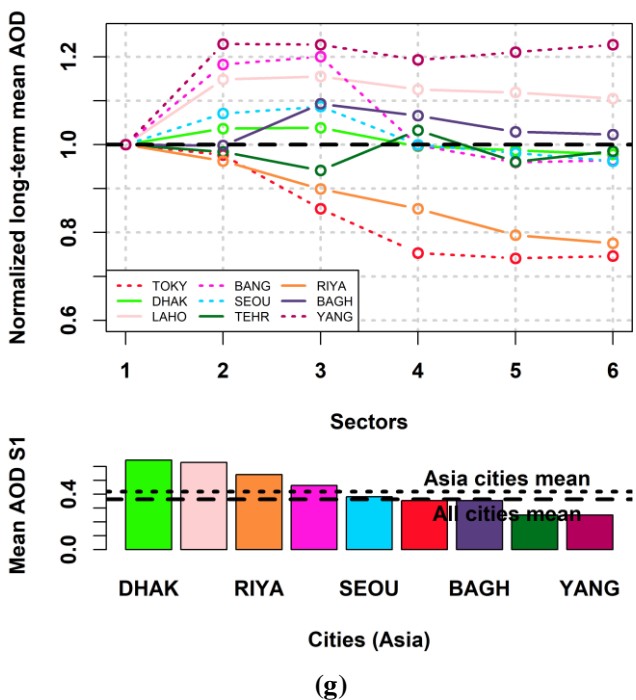

(g)

**Figure C1. Subplots a-g corresponds to deferent geographical domains. For every subplot: Upper panel: Long term mean AOD for the 6 different sectors normalized with the mean value of sector 1 (S1). The different colors correspond to different cities. Solid/dashed lines denotes inland/coastal cities, respectively. Lower panel: Long term mean AOD of S1 for every city in the geographical domain, denoted with the same colour used in the upper panel. The dashed line corresponds to the mean from all cities, while dashed lines gives the mean AOD for the cities of the corresponding geographical domain.**

AppendixD.

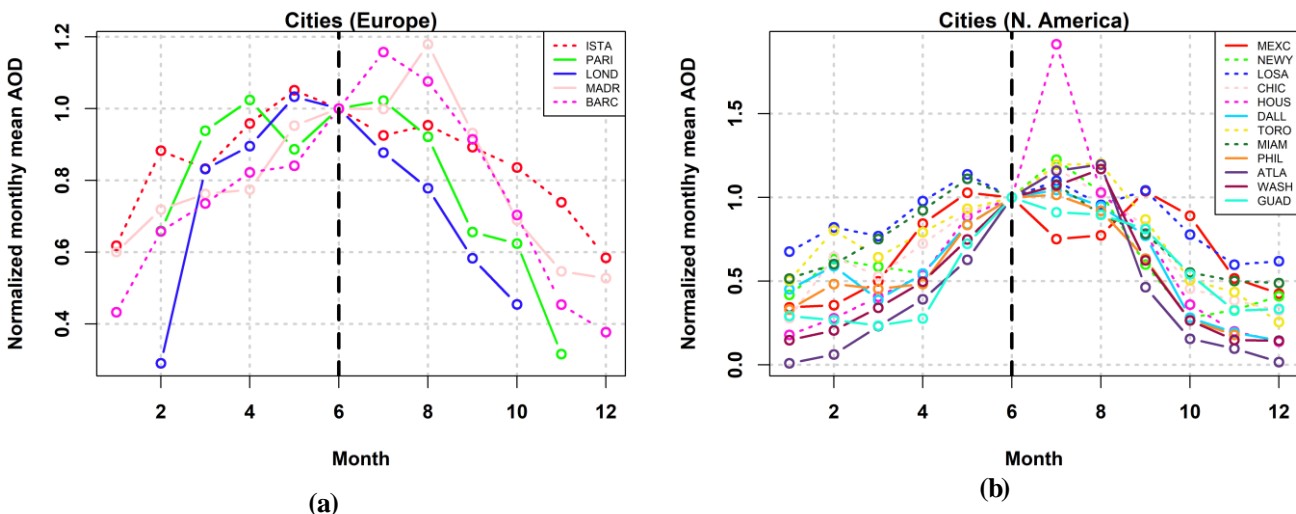

(a)                                                                      (b)





**(c)**

**(d)**

**(e)**

**(f)**



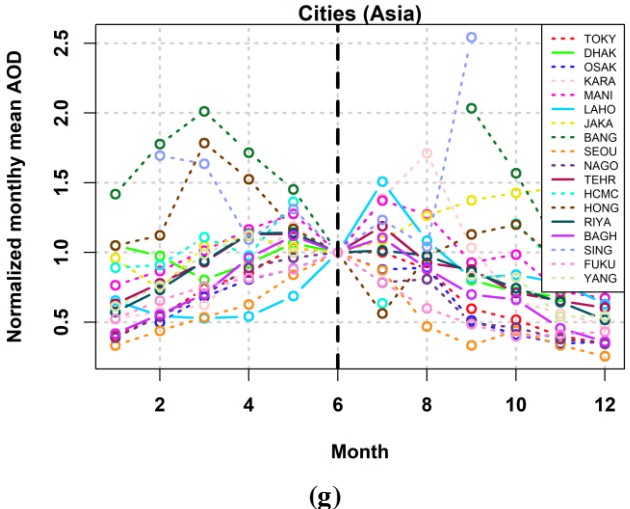

**(g)**

**Figure D1. Long term monthly mean AOD values (intra-annual variability) normalized against the mean value of month June. The different colors correspond to different cities. Solid/dashed lines denotes inland/coastal cities, respectively.**

Data availability. The MIDAS dataset (Gkikas et al., 2021, 2022) is available online at https://doi.org/10.5281/zenodo.4244106. AERONET observations were downloaded from the AERONET data base at https://aeronet.gsfc.nasa.gov. Population data are available online by the United Nations, Department of Economic and Social Affairs, Population Division at https://population.un.org/wup/Download/.

Author contributions. KP and SK designed the study and analysis. KP has analysed all satellite based AOD data and population data. IF and AG have contributed for the definition of the spatiotemporal use of the satellite data. AM and PR have performed the analysis of including the ground-based data. KP prepared the manuscript with contributions from all co-authors.

Acknowledgments. KP acknowledge support of this research by the Swiss Government Excellence Scholarship offered by the Swiss Government via the Federal Commision for Scholarships for Foreign Student FCS. SK, IF and KP would like to acknowledge the European Commission project EuroGEO e-shape (grant agreement No. 820852). AG was supported by the Hellenic Foundation for Research and Innovation (H.F.R.I.) under the "2nd Call for H.F.R.I. Research Projects to support Post-Doctoral Researchers" (project acronym: ATLANTAS, project number: 544). The MIDAS dataset has been developed in the framework of the DUST-GLASS project (grant No. 749461; European Union's Horizon 2020 Research and Innovation program under the Marie Skłodowska-Curie Actions). We acknowledge the data provided by AERONET and their continuous efforts in providing high-quality measurements and product.





Financial support. The publication of this article has been financed by the European Commission project EuroGEO e-shape (grant agreement No. 820852). This research has been supported by the Swiss Government Excellence Scholarship awarded to KP.

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
