# Peer review of "Aerosol optical depth regime over Megacities of the world"

_Atmospheric Chemistry and Physics, 2022_

## Author Comment (AC1)

**Reply to anonymous Referee #1**

We would like to thank anonymous referee#1 for his/her comments, that helped us improve the manuscript. We tried to address all reviewer's comments. In the following, analytical replies are provided to each of the reviewer's comments. Reviewer's comments are written in bold font. Line numbers, when provided refer to the new version with track changes.

**Summary**

**Large cities with large concentrations of population are facing serious environmental challenges. This study investigates the spatial and temporal characteristics of aerosol states in megacities using up-to-date spaceborne aerosol retrievals (AOD and Dust Optical Depth - DOD) with fine spatial resolution (0.1°x 0.1°). The author found that the spatial and temporal variability of AOD is closely related to the population agglomerations and the adoption of emission policies, which also is of significance for assessing the effectiveness of air pollution emission regulations.**

**I have some general and minor comments (please see below) which should be addressed prior to publication.**

**General comments:**

**Line 128-131: The author first mentioned the use of the MIDAS dataset with high resolution to analyze the spatial and temporal variability of AOD, and this dataset appears several times in the paper, so I think it is necessary to give specific details about MIDAS when it first appears.**

Reply

More details about the MIDAS datasets are given in the revised manuscript by replacing the previous paragraph with a new one (lines 125-137).

**As the author mentioned in the paper, a uniform pixel-based approach is applied for all cities. However, because different cities are located at different latitudes, increasing the uncertainty of pixel size in the east-west direction. Maybe the author could implement a distance conversion based on the relationship between latitude and pixel size to ensure that the city-centric division of space remains consistent? Based on the same thoughts, to reduce uncertainty, is it more reasonable to express the single measure of the AOD gradients in terms of distance as well?**

Reply

While the pixel size in north-south direction is the same ~ 110km for 10 pixels=1° for all cities, the pixel size differs for the east-west direction and the equivalent size of 10 pixels=1° for this direction has been derived for all cities and it is shown in the central panel of Fig. 4. As discussed in the manuscript for more than 75% of the cities the size in the east-west direction is 85-105km and for the 3 high latitude cities (~60° latitude) this size is ~65km.

In order to account for the differences in the pixel size for the east-west direction and ensure the consistency of the results among cities, we took for the high latitude cities London, Moscow and Saint Petersburg 2 extra pixels in the east-west direction for the construction of the time series of the urban and surroundings domains. Comparing the results before and after this adjustment for these 3 cities, it was found that the small increase in the data availability (~10%) wasn't enough to give trend results,

since for those cities the temporal criteria wasn't met. Additionally, this adjustment gave a 1-5% difference in the mean AOD values of those cities, difference which is smaller than the measuring uncertainty. Those findings led as to the conclusion that the uncertainties that are related to the differences in the pixel size could be considered as relatively low compared to those related to the data availability (see lines 220-224).

For the 6 sector analysis, according to reviewer suggestions, we expressed the single measure of the AOD gradients in terms of distance, in order to account for the difference in the east-west direction, by performed the following analysis:

For the north-south direction we took the same distance for all cities 110km. For the east-west direction the conversion to distance using the latitude was used (Fig. 4 central panel) and a mean distance of the 1 degree was derived by taking the mean of those two values. For almost 50% of the cities this distance is between 100 and 105 km and for all cities between 90-110 km, apart from high latitude cities. As a next step the AOD gradients were expressed in terms of distance per 100km. The results are almost the same in special gradients expressed either in km or degrees (with absolute differences below 0.015 and within the +/-10% in relative terms).

Hence, we replaced Fig. 8 with the corresponding one expressed in Δ(AOD) per 100km according to reviewer suggestions, figure's caption and description in subsection 3.1.2.1 has been changed accordingly, but no other change has been made in subsection 3.1.2.1. Section 2.4 has been changed accordingly, in order to give the methodology details (lines 191-208).

**Line 191-195: Why did the author construct the daily AOD time-series for urban domain by calculating the AOD median rather than AOD mean?**

Reply

On timescale of days, AOD may be normally distributed or not (next figure), as it is demostrated for example for the area of Tokyo city in the following figure for one random day. Following the recomandation by Sayer and Knobelspiesse (2019) median was selected as an aggregation method in order to account for days when the AOD values are not normally distributed. In order to clarify this better in the revised manuscript, an addition has been made (lines 179-181) in section 2.4 when it is first discussed, instead of 2.5

[Figure]

**minor comments**

**Line 99: Replace "surface-based" with "ground-based".**

Reply

The replacement has been made.

**Line 206: Add title number "2.6", and "Intra-annual" should be replaced by "intra-annual".**

Reply

The number 2.6 has been added for the subsection and the upper case letter was replaced by the lower case.

**Line 217: Add title number "2.7", "Intra-annual" should be replaced by "intraannual".**

Reply

The number 2.7 has been added for the subsection and the change to "intraanual" has been made in this line and throughout the manuscript.

**Line 227: Replace the title number "2.6" with "2.8".**

Reply

The title number has been replaced and the corresponding changes has been made throughout the manuscript.

**Line 322: Punctuation is in red**

Reply

The red color of punctuation changed to black.

**Line 440: Note the order in which the abbreviations of Ground Based (GB) appear.**

Reply

The title of the subsection 2.3 has been changed to "Ground-based (GB) measurements" in order to introduce the abbreviation for the rest of the manuscript and the title of the subsection 3.4 in which the comment refers to has been changed accordingly to "Evaluation with GB measurements (AERONET)"

**Figure 17 and figure 18 Figures 17, 18 seem to lack the serial numbers "(a)" and "(b)".**

Reply

The serial numbers were added to Figures 17, 18 and figure captions has been changed accordingly.

**There is a spelling error in the header of table A1 and "Agglomiration" should be changed to "Agglomeration".**

Reply

The appropriate change has been made in the header of table A1.

[revised manuscript text omitted]

---

## Author Comment (AC2)

We acknowledge anonymous referee#2 for his/her very useful comments, that helped us improve the manuscript. We tried to address all reviewer's comments. In the following, analytical replies are provided to each of the reviewer's comments. Reviewer's comments are written in bold font. Line numbers, when provided refer to the version with track changes.

**The manuscript by Papachristopoulou et al., "Aerosol optical depth regime over Megacities of the world", presents a study to investigate the spatial and temporal variability of urban aerosol state of 81 cities with population over 5 million, relying on daily satellite-based aerosol optical depth data. The focus and objectives of this study were quite straightforward and, probably at least partly for that reason, no actual problem points or weaknesses could be found in the approach and analysis. Because of this, my comments are also general or specific but minor. But of course, it should be mentioned in favor of the manuscript that the description and analysis was quite thorough, and these results bring useful information about the spatial and temporal variability of aerosols in the context of Megacities. I consider the topic and results of this manuscript to fit the scope of ACP and think it could be published if the minor comments below are addressed.**

**Line 146, here you mention about the number of days or seasons required for seasonal or annual means. Nine days out of 30 or three seasons out of four sound like somewhat loose limit. In place where there is a strong seasonal variability, it likely has a significant impact in the monthly mean whether the nine days of the month are in the very beginning or in the end of the month. Could you describe how did you arrive at these limits, and whether you tested more strict ones and the influence of these in the results.**

Reply

The choice of these limits for the number of available days and seasons is a compromise between representativeness and data availability to our target locations (only megacities). In the case of quantifying the aerosol loads using remote sensing techniques, the clouds presence is a limiting factor for the data availability, hence the statistics versus the variability within the month/season are always two fighting sides. For satellite remote sensing the data availability is limited also by the one measurement per day.

In order to arrive at these limits a sensitivity analysis has been performed testing different limits. By applying more strict results, substantially fewer cities left and for those left the results were alike with limits finally chosen.

Here, I am providing the results of the sensitivity tests by increasing the limit for the number of days.

For domain statistics:

For this analysis, the data were at their original 0.1x0.1 spatial resolution. The following barplots are giving the availability of seasons (in relative frequency) for every city, for the different temporal criteria applied. The first test (upper panel) was for unfiltered values. But since it will be a problem if a seasonal mean value resulted from only one day, temporal filtering was applied. Applying at least 10% data availability (9 days per season or 3 per month), the availability of seasons is reduced substantially below 75%, and this reduction is even bigger for the more strict limits. Thus, 9days was the optimum limit and even with this limit significant spatial gaps resulted (e.g. Fig. 2 b, c for Tokyo megacity).

[Figure]

For urban domain trends:

By increasing the limit of 9days to 18days, from 11 cities that were finally excluded from the trend analysis due to limited data availability, it would be tripled (33 cities would be excluded).

[Figure]

Regarding the impact at the trends using the more strict limit of 18 days, this was negligible for the magnitude of the slope (at 3rd decimal point, next figure) and most importantly there were no difference at the sign of the trends.

[Figure]

In the revised version of the manuscript has been added that we arrived at those limits by conducting a sensitivity analysis (lines 159-160).

**Line 287, you write "... revealed that the 4x4 pixels area is a domain well fitted ...". How it was "revealed", could you describe your analysis/conclusions in somewhat more detail.**

Reply

More details were added to the subsection 3.1.1 of the revised manuscript (lines 261-269) of the analysis that led to the conclusion that the 4x4 pixel area is the domain that well represents the extend of urban agglomerations, namely the contiguous built up area for all cities.

**Line 345, your finding that the city of Los Angeles resulted in different trend than overall western US was interesting. How about the results in the Figure 16, did they perhaps give a similar indication that also in the scale of city center vs surrounding area, a similar difference in the trend was found? Related to Los Angeles (LOSA), is there a reason why it was not included in the Figure C1b?**

Reply

In general, individual geographical aspects of each city analyzed was difficult to be included in the manuscript due to the large number of included cities. Trying to group cities geographically was an aspect that does not describe all the individual details, as is the case of Los Angeles.

The fact that we found the sign of the trend for Los Angeles opposite to the sign that other studies report for the same area, but with coarser spatial resolution of the satellite retrievals, is indeed something interesting to mention and explain further.

Regarding Figure 16, it demonstrates the differences that were found for the trends in the scale of city center vs surrounding area, but the reported differences were only in magnitude. For no city there was difference in the sign of the trend between city center and the surrounding area.

In the Figure C1 and in general for the spatial (six sector) analysis (see Sect. 3.1.2.1) Major Metropolitan Areas (M.M.A.s), like Los Angeles, were excluded, along with the high altitudes cities (see also lines 118-119). Our choice was based on the fact that the results of the AOD gradients for those cities are affected by other factors. For example, Los Angeles it is actual the M.M.A. including Long Beach and Santa Ana (see Table A1). This means that up to sector 6 around Los Angeles city center are including other great cities, which is out of the scope of the six-sector analysis.

**Line 410, I would think that the large interannual variability in North America, most notably in Atlanta is related to the significant impact by biogenic aerosols, discussed in many papers (one of the best known is Goldstein et al. 2009). Could you elaborate on this matter.**

**REFERENCES**

**Goldstein, A. H., Koven, C. D., Heald, C. L., and Fung, I. Y.: Biogenic carbon and anthropogenic pollutants combine to form a cooling haze over the southeastern United States, P. Natl. Acad. Sci. USA, 106, 8835–8840, 2009.**

Reply

We are thankful to the reviewer for his comment and suggestion for this subject. The discussion about the impact of biogenic aerosols in the large intrannual variability in southeastern US was added in the revised version of the manuscript (lines 442-447).

[revised manuscript text omitted]